# Use of reflected GNSS SNR data to retrieve either soil moisture or vegetation height over a wheat crop

Sibo Zhang[1,2], Nicolas Roussel[3], Karen Boniface[2,3,4,6], Minh Cuong Ha[3], Frédéric Frappart[3,5], José Darrozes[3], Frédéric Baup[4], and Jean-Christophe Calvet[1]

5   [1]CNRM – UMR3589 (Météo-France, CNRS), Toulouse, France

[2]Fondation STAE, Toulouse, France

[3]GET (UMR5563 CNRS/Université Paul Sabatier, UR254 IRD), Toulouse, France

[4]CESBIO, Université de Toulouse, CNES/CNRS/IRD/UPS, Toulouse, France

[5]LEGOS – UMR566 (CNES, CNRS, IRD, UPS), Toulouse, France

10   [6]now at Joint Research Centre / European Commission, Ispra, Italy

*Correspondence to*: Jean-Christophe Calvet (jean-christophe.calvet@meteo.fr)

**Abstract.** This work aims to estimate soil moisture and vegetation height from Global Navigation Satellite System (GNSS) Signal to Noise Ratio (SNR) data using direct and reflected signals by the land surface surrounding a ground-based antenna. Observations are collected over a rainfed wheat field in southwestern France. Surface soil moisture is retrieved based on 15 SNR phases estimated by the Least Square Estimation method, assuming the relative antenna height is constant. It is found that vegetation growth breaks up the constant relative antenna height assumption. A vegetation height retrieval algorithm is proposed using the SNR dominant period (the peak period in the average power spectrum derived from a wavelet analysis of SNR). Soil moisture and vegetation height are retrieved at different time periods (before and after vegetation significant growth in March, respectively). The retrievals are compared with two independent reference datasets: *in situ* observations of 20 soil moisture and vegetation height, and numerical simulations of soil moisture, vegetation height and above-ground dry biomass from the ISBA (Interactions between Soil, Biosphere and Atmosphere) land surface model. Results show that changes in soil moisture mainly affect the multipath phase of the SNR data (assuming the relative antenna height is constant) with little change in the dominant period of the SNR data, whereas changes in vegetation height are more likely to modulate the SNR dominant period. Surface volumetric soil moisture can be estimated ($R^2 = 0.74$, RMSE = 0.009 $m^3 m^{-3}$) when the 25 wheat is smaller than one wavelength (~ 19 cm). The quality of the estimates markedly decreases when the vegetation height increases. This is because the reflected GNSS signal is less affected by the soil. When vegetation replaces soil as the dominant reflecting surface, a wavelet analysis provides an accurate estimation of the wheat crop height ($R^2 = 0.98$, RMSE = 6.2 cm). The latter correlates with modeled above-ground dry biomass of the wheat from stem elongation to ripening. It is found that the vegetation height retrievals are sensitive to changes in plant height of at least one wavelength. A simple 30 smoothing of the retrieved plant height allows an excellent matching to *in situ* observations, and to modeled above-ground dry biomass.

# 1 Introduction

*In situ* observations of soil moisture and vegetation variables are key to validate land surface models and satellite-derived products. Recent international initiatives, such as the International Soil Moisture Network (Dorigo et al., 2013) or the Committee on Earth Observation Satellites (CEOS) Land Product Validation group (Morisette et al., 2006) have improved the access to such observations. However, they remain very sparse and there is a need to develop new automatic techniques to monitor land surface variables at a local scale. Global Navigation Satellite System (GNSS) reflectometry could be a solution. A number of studies demonstrated that GNSS multipath signals can be used to retrieve various geophysical parameters of the surface surrounding a GNSS receiving antenna (Motte et al., 2016). Over land, variables such as soil moisture, snow depth and vegetation status can be observed (Larson et al., 2008; Small et al., 2010; Larson and Nievinski, 2013; Wan et al., 2015; Boniface et al., 2015; Larson, 2016; Roussel et al., 2016). GNSS satellites operate at the L-band microwave frequency domain (between 1.2 GHz and 1.6 GHz). At these relatively low frequencies, the microwave signal is less perturbed by atmospheric effects and can better penetrate clouds and heavy rains than higher frequency signals. This ensures continuous operations, in all weather conditions, at either daytime or nighttime. The L-band signal emitted or reflected by terrestrial surfaces is related to surface parameters like surface soil moisture, roughness or vegetation characteristics. These properties have been exploited by e.g. the Soil Moisture and Ocean Salinity (SMOS) satellite and the Soil Moisture Active Passive (SMAP) missions (Kerr et al., 2001; Chan et al., 2016) for Earth surface remote sensing applications. While SMOS is a radiometer and measures the Earth surface microwave emission (passive microwaves), GNSS satellites emit a radar signal (active microwaves). Active microwaves can present improved temporal and spatial resolutions, but the signal may be more sensitive to the structure of the surface, such as soil roughness or vegetation effects than for passive microwaves (Wigneron et al., 1999; Njoku et al., 2002).

Existing geodetic-quality GNSS networks have the potential to provide a large number of *in situ* observations, depending on the receiver technology: (1) waveform acquisition with a specific receiver using two antennas (one zenith-oriented antenna and one surface-oriented antenna), called GNSS reflectometry (GNSS-R) technique (Zavarotny et al., 2014) or (2) GNSS signal strength represented by the Signal-to-Noise Ratio (SNR) acquired with a classical geodetic receiver using one antenna, called SNR GNSS interferometric reflectometry (GNSS-IR) technique (Larson, 2016). GNSS networks can be used to monitor small or large areas depending on the antenna height and satellite elevation (Roussel et al., 2014). Continuous monitoring of surface soil moisture can be made over a long period at spatial scales ranging from 100 m$^2$ (antenna height of about 2 m) to 8000 m$^2$ (antenna height of about 150 m) for classical geodetic receiver but can reach a few thousand square kilometers with waveform receivers embedded on satellites (e.g. TechDemoSat-1 mission, Foti et al. (2015)).

Using the SNR GNSS-IR technique, Larson et al. (2008) showed that SNR data obtained from existing networks of single ground-based geodetic antennas can be used to infer soil moisture. Other GNSS methods (besides reflectometry) can be used. For example, Koch et al. (2016) used three geodetic GNSS antennas (one was installed above the soil, the other two

were buried at a depth of 10 cm), to measure the GNSS signal strength attenuation and to retrieve soil moisture over bare soil.

A network called PBO $H_2O$ based on single GNSS antennas at Plate Boundary Observatory (PBO) sites is currently used in western regions of the USA to monitor surface soil moisture (Larson et al., 2013; Chew et al., 2016) and snow depth (Larson and Nievinski, 2013; Boniface et al., 2015). It must be noted that most of the 161 GNSS stations of this network are located in mountainous areas or in areas of California characterized by a relatively arid climate. They are surrounded by sparse vegetation and are therefore not adapted to vegetation growth studies.

In the SNR GNSS-IR technique, the interference between the direct and the reflected signals is observed through temporal variations of the SNR data (Bilich and Larson 2007; Zavorotny et al., 2010; Chew et al., 2014). Changes in geophysical or biophysical parameters affect the phase, amplitude and frequency of the SNR modulation pattern. The SNR is also influenced by surface roughness and by the position of the antenna with respect to the surface and to the satellite (Larson and Nievinski 2013; Chew et al., 2016). The SNR modulation primarily depends on:

- the relative height of the GNSS antenna above the reflecting surface (ground or vegetation surface),
- satellite elevation,
- the superposition of the direct signal and of the reflected signal, which varies along with changes in the satellite track positions,
- Right Hand Circular Polarization (RHCP) and Left Hand Circular Polarization (LHCP) gain pattern of the receiving antenna, (RHCP usually increases the SNR when the satellite elevation angle increases, LHCP is related to imperfections of the antenna and is greater than RHCP for the reflected signal);
- reflection coefficients for the reflecting surface, related to the water content and to the ground mineralogical content of the reflecting surface,
- surface topography and roughness and
- the satellite transmitted power.

A soil moisture retrieval algorithm from SNR data was derived by Chew et al. (2014) over bare soil. In subsequent modeling studies Chew et al. (2015) showed that the vegetation canopies affected the SNR modulation pattern. They showed that vegetation growth tended to trigger a decrease of the SNR amplitude. Because the vegetation effects tended to perturb the soil moisture retrieval, Chew et al. (2016) proposed an improved algorithm for soil moisture retrieval in vegetated environments, which used the amplitude decrease extent to decide when vegetation influence was too large. They used a model database for the SNR of L2C signal to remove most significant vegetation effects for the sites they considered in Western USA. Small et al. (2016) further compared different algorithms of GNSS-IR soil moisture retrieval in the presence of vegetation. Roussel et al. (2016) integrated both GPS and GLONASS SNR data to retrieve soil moisture over bare soil. Using data from a field study, Wan et al. (2015) showed that the amplitude of the SNR data presented a good linear relationship with the vegetation water content (VWC), but it was restricted to VWC values of less than ~1 kg m$^{-2}$. In addition

to the amplitude of the SNR data, it was also possible to infer VWC by the $MP1_{rms}$ index, which is a linear combination of L1 and L2 carrier phase data and L1 pseudorange data (Small et al., 2010), and by the NMRI (Normalized Microwave Reflection Index) which is derived from the $MP1_{rms}$ (Small et al., 2014; Larson and Small 2014).

In this study, the SNR GNSS-IR technique was used to analyze GNSS SNR data obtained with a single classical geodetic antenna receiver over an intensively cultivated wheat field in southwestern France. The data were used to retrieve either soil moisture or relative vegetation height during the growing period of the wheat crop. The method proposed by Chew et al. (2016) (hereafter referred to as CH16) was used to retrieve soil moisture. Moreover, we performed a wavelet analysis in order to extract the dominant period of the SNR. We investigated to what extent vegetation height influenced the dominant period resulting from the wavelet analysis. The main justification for investigating the impact of vegetation height was that it impacted the relative antenna height (the distance from the antenna to the reflecting surface). Vegetation growth tended to decrease the relative antenna height and broke up the constant height assumption used in soil moisture retrieval algorithms. In this context, key objectives of this study were to (1) assess the soil moisture retrieval technique in either low or tall vegetation conditions, and (2) retrieve vegetation height along the wheat growth cycle.

## 2 Materials and methods

### 2.1. SNR data and pre-processing

The GNSS SNR data were acquired from an antenna at 2.51 m above the soil surface over an experimental field covered by rainfed winter wheat in Lamasquère, France (43°29'10"N, 1°13'57"E, see Fig. S1 in the Supplement). These GNSS data were collected by GET (Géosciences Environnement Toulouse) for a whole growing season, from January to July 2015. A Leica GR25 receiver equipped with an AS10 antenna was used and data were acquired at a sample frequency rate of 1 Hz. Only the S1C SNR signal strength on the civilian L1 C/A channel of the GPS constellation was used in this study because the used receiver could not track the L2C signal. The latter is only transmitted by the recent Block IIR-M ("Replenishment Modernized") and IIF ("Follow-on") GPS satellites. Vey et al. (2016) showed that soil moisture root mean square difference between L2C and L1 was 0.03 $m^3m^{-3}$. The quality of the more recently available L2C signal (used by PBO $H_2O$ (CH16)) is higher than either L1 C/A or L2P from non-code tracking receivers. However, a number of studies (e.g. Vey et al., 2016) showed that the SNR of the L1 C/A signal can be used to provide reliable soil moisture estimates over sparse vegetation and bare soil surface, although it is less precise than the L2C signal. Although data from other constellations were also acquired (e.g., GLONASS, GALILEO), their orbital parameters such as satellite track positions or satellite altitude were not the same. In order to be consistent with the GPS-only studies of Larson et al. (2008), CH16, and Small et al. (2016), we only used GPS SNR data. For our site, four GPS satellites out of 32 were excluded from the analysis because their data were incomplete (GPS03, 20, 26, these numbers corresponding to their Pseudo-Random Noise (PRN) numbers) or not received (GPS08). Finally, GPS SNR data were missing for only nine days: 8 and 9 February, 3 April and from 13 to 18 May 2015.

Following the method proposed by Larson et al. (2010), a low-order polynomial was fit to the SNR data, and the modulation pattern was then derived from the SNR by subtracting this polynomial from the SNR data. The logarithmic dB-Hz units were converted to a linear scale in V V$^{-1}$ using the following conversion equation: $SNR_{linear} = 10^{\frac{SNR}{20}}$ (Vey et al., 2016). Figure 1a shows an example of the detrended multipath SNR data for the ascending track of GPS01 on 21 January 2015. The periodic signature of the multipath SNR data is visible. We only analyzed the modulation patterns in a valid segment for satellite configurations corresponding to low elevation angles, ranging from 5 to 20 degrees. This corresponded to a valid segment data recording of less than one hour (40 to 50 minutes). We excluded very low elevation angles (less than 5 degrees) in order to avoid spurious effects from trees and artificial surfaces surrounding the field. Because the SNR signal amplitude was much reduced and the wave pattern was not visible at high elevations for our field observations, we excluded elevation angles larger than 20 degrees.

## 2.2. Soil moisture and vegetation characteristics

The field campaign was part of a coordinated effort led by CESBIO (Centre d'Etudes Spatiales de la BIOsphère) to monitor crops in southwestern France using both *in situ* and satellite Earth Observation data. Independent *in situ* observations of soil moisture and vegetation height were made together with model simulations of these quantities. Both observations and simulations were used to validate soil moisture and vegetation height retrievals.

Since the whole wheat growing cycle was examined, both soil moisture and vegetation modulated the multipath SNR pattern. Soil roughness was considered as stable in time from sowing to harvest. Soil in the close vicinity of the antenna consisted of 18% of sand, 41% of clay, and 41% of silt. The row spacing of the wheat crop was 15 cm.

The wheat was sown during the autumn, on 1 October 2014 and was harvested from 26 to 30 June 2015. Volumetric soil moisture (VSM) was measured by FDR (Frequency Domain Reflectometry) ML2 Thetaprobes and was continuously monitored at a depth of 5 cm from 16 January to 10 March 2015 and from 30 March to 26 May 2015. Measurements of crop height were performed at seven dates during the plant growing cycle. The canopy height did not exceed 0.1 m at wintertime and rapidly increased at springtime: it reached 0.2 m on 10 March 2015 and 1 m on 29 May. It dropped to 0.39 m on 18 June because of a lodging event. The exact date of lodging could not be precisely determined. It could be inferred that lodging happened between 29 May and 18 June.

In addition to *in situ* observations, simulations of surface soil moisture (0-10 cm top soil layer), plant height and above-ground dry biomass were performed for this site by CNRM (Centre National de Recherches Météorologiques) using the ISBA (Interactions between Soil, Biosphere, and Atmosphere) land surface model within the SURFEX (version 8.0) modeling platform (Masson et al., 2013). The ISBA configuration and the atmospheric analysis used to force the model are described in Lafont et al. (2012). The C3 crop plant functioning type and a multilayer representation of the soil hydrology are considered. The model soil depth is 12 meters, with 15 layers and the layer thickness increases from the top surface layer

to the deepest layers (Decharme et al., 2011). These simulations were used as an independent benchmark for soil moisture and vegetation variables.

## 2.3. Multipath SNR characteristics

Due to the motion of the GPS satellites, the path delay between the direct and reflected signals causes an interference pattern in the signal power of SNR data. The distance from the antenna to the dominant reflecting surface directly affects the SNR frequency/period.

As noted by Georgiadou and Kleusberg (1988) and Bilich and Larson (2007), assuming the ground surface is horizontal, the additional distance ($\delta$) travelled by a reflected signal relative to the direct signal is

$$\delta = 2h\sin(\theta) \tag{1}$$

where $h$ is the relative antenna height, and $\theta$ is the satellite elevation angle. This path delay $\delta$ can also be expressed in terms of the multipath relative phase $\psi$ :

$$\psi = 2\pi\frac{\delta}{\lambda} \tag{2}$$

where $\lambda$ represents the L1 wavelength (0.1903 m).

Thus the multipath frequency ($f$) and period ($T$) can be written as:

$$\omega = 2\pi f = \frac{d\psi}{dt} = \frac{4\pi}{\lambda}h\cos(\theta)\frac{d\theta}{dt} \tag{3}$$

$$\frac{1}{T} = f = \frac{2h\cos(\theta)}{\lambda}\frac{d\theta}{dt} \tag{4}$$

This means that the relative antenna height ($h$) directly affects multipath frequency $f$ and period $T$. Antennas far above the reflecting surface have higher multipath frequencies (smaller multipath periods) than antennas closer to the reflecting surface. Furthermore, satellite geometric information and motion substantially influences $T$ due to the $\cos(\theta)$ and $d\theta/dt$ terms in equation (4). When satellite passes reach high elevation angles $d\theta/dt$ becomes larger (Bilich and Larson, 2007). Conversely, satellites with passes presenting small maximum elevations present smaller $d\theta/dt$ values than satellites orbiting overhead. Contrasting configurations are illustrated in the Supplement (Fig. S2). In order to limit the impact of these differences from satellite motion, only the full-track data with at least 40 degree maximum elevation angle were selected. Among the remaining tracks we removed the slowly moving tracks whose maximum $\cos\theta \cdot d\theta/dt$ was less than $9.5\times10^{-5}$ rad s⁻¹ (threshold value based on our field observations) of the valid segment (elevation angles ranging between 5 and 20 degrees). This specific data sorting was only made for vegetation height retrieval (Sect. 2.5). After this selection, the number of available satellite tracks was 37 per day.

Provided the reflecting surface is stable, the a priori antenna height can be used to estimate the SNR frequency. The SNR frequency is used to calculate the multipath SNR phase, and then the SNR phase is used to estimate VSM (Sect. 2.4). If the reflecting surface is changing in response to vegetation growth, relative vegetation height can be retrieved instead of VSM by directly estimating the dynamic SNR frequency/period with a wavelet analysis (Sect. 2.5).

## 2.4. Soil moisture retrieval

As the SNR frequency is known (Eq. (4)), it is possible to estimate the SNR amplitude and phase. Larson et al. (2008) and Larson et al. (2010) showed that phase varies linearly with VSM in $m^3 m^{-3}$ ($R^2$ = 0.76 to 0.90). Retrieving absolute VSM values in $m^3 m^{-3}$ is possible after a calibration phase. This result was used by Chew et al. (2014) to develop an algorithm to estimate surface soil moisture (top 5 cm) over bare soil.

For bare soil, changes in surface soil moisture affect the signal penetration depth. The latter can be very small in wet conditions and tends to increase in dry conditions, up to a few centimeters (Chew et al., 2014; Roussel et al., 2016). This is a small change with respect to the antenna height (2.51 m in this study). Consequently, the relative antenna height ($h$) is considered as a constant ($h_c$ = 2.51m) in this Section. Using sine of the elevation angle ($\sin(\theta)$) as the independent variable, the modulation frequency becomes proportional to $h_c$. Then the multipath SNR can be expressed as (Larson et al., 2008):

$$SNR_{mpi} = A\cos(\frac{4\pi h_c}{\lambda}\sin(\theta) + \varphi_{mpi}) \tag{5}$$

The least square estimation (LSE) method proposed by Larson et al. (2008) is used to estimate the multipath amplitude ($A$) and multipath phase ($\varphi_{mpi}$) from the multipath SNR data. Then, $\varphi_{mpi}$ can be used to estimate the soil moisture changes (CH16),

$$VSM_t = S \cdot \Delta\varphi_t + VSM_{resid} \tag{6}$$

Phase changes $\Delta\varphi_t = \varphi_t - \varphi_0$ are calculated with respect to $\varphi_0$, the reference phase. We used the method proposed by CH16 consisting in estimating $\varphi_0$ as the mean of the lowest 15% of the $\varphi_{mpi}$ data for each track during the retrieval period. The same condition was used to estimate the *VSM_resid* residual soil moisture from the *in situ* VSM observations. The *VSM_resid* was taken as the minimum soil moisture observation, which presented a value of 0.252 $m^3 m^{-3}$ during the retrieval period. The $S$ parameter (in $m^3 m^{-3}degree^{-1}$) is the slope of the linear relationship between phase changes and soil moisture. For time series with no significant vegetation effects, $S$ = 0.0148 $m^3 m^{-3}degree^{-1}$ for L2C signal (CH16). Following CH16, the median soil moisture estimate from all available satellite tracks (66 per day) that passed at different times during the day was used as the final soil moisture estimate.

We also used the *in situ* $VSM_t$, $\Delta\varphi_t$ and $VSM_{resid}$ to fit a locally adjusted slope. The retrieval of the $S$ parameter requires at least one or two months of VSM *in situ* observations because soil moisture conditions ranging from dry to wet need to be sampled. However, if a scaled soil wetness index is used instead of soil moisture, no *in situ* VSM observations are needed. Alternatively, the phase time series can be normalized for each satellite track, and using $S$ is not needed. We considered the median value of the normalized phases from all available satellite tracks (66 per day) as the final scaled soil wetness index ($\varphi_{index}$) for each day:

$$\varphi_{index} = \frac{\varphi - \varphi_{\min}}{\varphi_{\max} - \varphi_{\min}} \tag{7}$$

VSM could then be estimated from $\varphi_{index}$:

$$VSM = VSM_{obs\_\min} + \varphi_{index} \cdot (VSM_{obs\_\max} - VSM_{obs\_\min}) \tag{8}$$

$VSM_{obs\_min}$ and $VSM_{obs\_max}$ are the minimum and maximum *in situ* VSM observations during the experimental time period, respectively.

CH16 defined the normalized amplitude ($A_{norm}$) as the ratio of amplitude to the average of the top 20 % amplitude values. The $A_{norm}$ time series can be used to assess whether or not vegetation effects are significant. Values of $A_{norm}$ above 0.78 (dimensionless) indicate that vegetation effects are small (CH16). In conditions of significant vegetation effects CH16 used an algorithm able to correct the phase for vegetation effects. This algorithm is based on an unpublished lookup table. Since we were not able to correct for vegetation effects, we retrieved surface soil moisture during a period with rather sparse vegetation, from 16 January to 5 March. During this time span, $A_{norm}$ was above 0.78 as shown in Fig. 2 (black dots).

**2.5. Vegetation height retrieval using a wavelet analysis**

While vegetation grows, the vegetation surface gradually replaces the bare soil surface as the dominant reflecting surface. As a consequence, the height ($h$) of the antenna above the reflecting surface decreases. Equation (4) shows that changes in $h$ impact $T$. This property allows the use of changes in $T$ values to infer changes in $h$, and further estimate relative vegetation height. To retrieve relative vegetation height we propose a new approach based on wavelet analysis. Wavelets have been used for many years in signal processing studies in geosciences (Ouillon et al., 1995; Darrozes et al., 1997; Gaillot et al., 1999), astrophysics (Escalera and MacGillivray, 1995), meteorology (e.g. Hagelberg and Helland, 1995; Torrence and Compo, 1998), hydrology (Labat, 2005) and in many other fields. The wavelet analysis is well suited for analyzing time series with non-stationary power and frequency changes across time as illustrated by Fig. 1. Our wavelet analysis methodology is based on the WaveletComp R-package (Roesch et al., 2014). To analyze the period structure, we used a

well-known Morlet mother function which comes from a combination of a Gaussian function and a sinusoidal function (Fig. S3 in the Supplement). Due to its shape, Morlet daughters allow detection of singularities in all scales/periods of the spectrum. Morlet wavelet is also well suited for environmental analysis (Grinsted et al., 2004). We calculate the Morlet wavelet transform of the multipath SNR and evaluate the power spectrum of the multipath SNR signal (see Eqs. S1-S4 in the
Supplement).

Vegetation height can be retrieved using the dominant SNR period ($T_d$), which is the peak period of the average power spectrum derived from a wavelet analysis of SNR, from the multipath SNR segment at elevation angles from 5 to 20 degrees. After obtaining $T_d$ time series, the relative antenna height ($h$) can be derived from Eq. (4) as:

$$h = \frac{\lambda}{2\cos\theta_{E9} \cdot \dfrac{d\theta_{E9}}{dt} \cdot T_d} \tag{9}$$

The $T_d$ value is used to represent the multipath SNR data in order to estimate $h$. Also, changes in the elevation angle ($\theta$) and in $d\theta/dt$ have to be accounted for. In this study, changes in $h$ were surveyed across dates at an elevation angle of 9 degree (See Sect. 3.2).

Changes in relative antenna height ($h$) during vegetation growth are directly related to vegetation height increase:

$$\Delta H = h_0 - h \tag{10}$$

Similarly to the phase change estimates ($\Delta\varphi_t$ in Sect. 2.4), $h_0$ is the median value of the top 15% $h$ data during the whole wheat growth cycle for each track.

The final retrieved vegetation height ($H$) is based on the mean relative antenna height change from all available satellite tracks ($N = 37$), plus one wavelength:

$$H = \frac{\displaystyle\sum_N \Delta H}{N} + \lambda \tag{11}$$

The minimum value of $H$ is one wavelength. Therefore Eq. (11) can only be applied when the wheat height is higher than one wavelength (0.19 m for L1).

It must be noted that it is not necessary to retrieve soil moisture before retrieving vegetation height.

## 2.6. GDD (growing degree days) model

Because of the lack of *in situ* records of the field wheat growth stages, we built a reference GDD model based on the wheat
growth stage dates observed at the same location in 2010 (Duveiller et al., 2011; Fieuzal et al., 2013, Betbeder et al., 2016). The GDD model is described in the Supplement (Eqs. S5-S6 and Fig. S4).

## 3. Results

### 3.1 Soil moisture retrieval

Figure 3 presents the surface soil moisture retrievals from 16 January to 5 March 2015, together with independent *in situ* VSM observations and ISBA simulations. The VSM retrievals are derived from GPS SNR observations using Eq. (6) in
sparse vegetation conditions, when $A_{norm}$ is above 0.78, with the a priori $S$ value of 0.0148 $m^3m^{-3}degree^{-1}$ (Fig. 3a) and the adjusted local slope $S$ = 0.0033 $m^3m^{-3}degree^{-1}$ (Fig. 3b). This adjusted $S$ value is the mean of slope values obtained for satellite tracks whose phase presented a linear correlation with *in situ* soil moisture higher than 0.9. This occurred for the ascending tracks of GPS 13, 21, 24 and 30 and for the descending tracks of GPS 05, 09, 10, 15, and 23. Figure 3c shows the VSM retrievals from the scaled soil wetness index (Eq. (8)).

The GPS and ISBA scores are given in Table 1. The mean soil moisture values during the experimental period are 0.27, 0.28, 0.31, 0.26, and 0.28 $m^3m^{-3}$ for *in situ* VSM measurements, ISBA simulations, GPS retrievals with $S$ = 0.0148 $m^3m^{-3}degree^{-1}$, GPS retrievals with $S$ = 0.0033 $m^3m^{-3}degree^{-1}$, and GPS retrievals from the scaled soil wetness index, respectively.

In Fig. 3, the sub-daily statistical distribution of the VSM retrievals is indicated by box plots. The range of daily standard deviation value of the various VSM estimates is shown in Table 2. The *in situ* VSM measurements present the smallest sub-
daily variability, with a mean standard deviation value of 0.002 $m^3m^{-3}$. The largest variability is obtained for the GPS retrievals based on the a priori slope value $S$ = 0.0148 $m^3m^{-3}degree^{-1}$, with a mean standard deviation value of 0.036 $m^3m^{-3}$. GPS retrievals based on the adjusted slope value $S$ = 0.0033 $m^3m^{-3}degree^{-1}$ presents intermediate values (0.008 $m^3m^{-3}$), together with those based on the scaled soil wetness index (0.009 $m^3m^{-3}$) and with the ISBA simulations (0.005 $m^3m^{-3}$). Figure 3 shows that the sub-daily variability of GPS VSM retrievals tends to increase during the last 10 days of the retrieval
period.

It must be noted that GPS data are missing on 8 and 9 February, and that the ISBA simulations indicate soil freezing (i.e. the presence of ice in the top soil layer) from 4 to 9 February. This period was excluded from the comparison. In the end, there were 47 valid observation days for the statistical analysis of the retrieved surface VSM, among which 43 days could be compared with model simulations.

The GPS VSM daily mean retrievals based on the CH16 method present a good agreement with both *in situ* observations and ISBA simulations: MAE (Mean Absolute Error) and RMSE (Root Mean Square Error) are lower than 0.05 $m^3m^{-3}$, and SDD (Standard Deviation of Differences) does not exceed 0.04 $m^3m^{-3}$ (Table 1). The errors are reduced by at least 50 % when the local adjusted slope is used. When the scaled soil wetness index is used, the errors are further reduced.

Figure 4a and 4b show the retrieved soil moisture as a function of the *in situ* observations for a priori and adjusted slopes ($S$
= 0.0148 $m^3m^{-3}degree^{-1}$ and $S$ = 0.0033 $m^3m^{-3}degree^{-1}$, respectively) from all available satellite tracks (66 per day), not only those tracks used for fitting the slope (see Supplement Fig. S5). The corresponding improvements in score values are given in Table 1: the MAE decreases from 0.036 to 0.011 $m^3m^{-3}$, the RMSE decreases from 0.046 to 0.014 $m^3m^{-3}$, the SDD

decreases from 0.036 to 0.009 $m^3m^{-3}$. The retrievals based on the a priori slope markedly overestimate VSM in wet conditions. On the other hand, the retrievals based on the adjusted slope only slightly underestimate VSM. This shows that adjusting the slope is critical and has a major impact on the retrieval accuracy. Furthermore, Figure 4c gives the retrievals based on the scaled soil wetness index. Scores are further improved: the MAE decreases to 0.007 $m^3m^{-3}$, RMSE to 0.009 $m^3m^{-3}$, and SDD to 0.008 $m^3m^{-3}$.

We also compared the retrievals with the independent ISBA simulations. The ISBA model VSM simulations present a better agreement with the *in situ* VSM observations than the GPS retrievals, for all the scores, as shown by Table 1 (last column) and Fig. 3. In particular, $R^2 = 0.88$ for ISBA simulations, against $R^2 = 0.74$ for GPS retrievals. This shows that the ISBA simulations can be used as a reference to assess local GPS retrievals for this site. The statistical scores resulting from the comparison between the GPS retrievals and the simulations are similar to those based on *in situ* observations.

After 5 March, $A_{norm}$ drops below 0.78 (Fig. 2), and the VSM retrievals are not valid. We made an attempt to retrieve VSM from 6 to 15 March. We obtained 10 VSM retrieved values and we compared them with ISBA VSM simulations, because *in situ* observations were lacking. The $R^2$ score decreased from 0.63 before 6 March (Table 1) to only 0.21 from 6 to 15 March. This result confirms that the empirical $A_{norm}$ threshold (0.78) is a good way to assess the VSM retrieval feasibility over vegetated areas. Additionally, we found that adjusting the $A_{norm}$ threshold from 0.78 to 0.88 permitted making a distinction between harvest and post-harvest (after 30 June) $A_{norm}$ values in Fig. 2. Four more days (2-5 March) are excluded. Figure 3 shows that the 25-75% percentile intervals for these days are larger, but the maximum retrieval differences for these days are acceptable, around 0.03 $m^3m^{-3}$.

### 3.2 Dominant SNR period analysis during the wheat growth cycle

Figure 1 shows an example of the multipath SNR data from the ascending track of GPS01 on 21 January 2015. Its average power spectrum (Fig. 1b) derived from a wavelet analysis is also shown, together with the power spectrum (Fig. 1c) for periods ranging from 128 to 1024 s. The average power spectrum presents a single peak and the corresponding peak period is 362 s. The SNR data is reconstructed well (red line in Fig. 1a), using this peak period. Both phases and amplitudes match very well. This shows that the peak period from the average power spectrum can be used to represent the multipath SNR data. Limiting elevation angle values from 5 to 20 degrees (Sect. 2.1) ensures a relatively stable value of the peak period. The peak period is considered as the dominant period ($T_d$) of the multipath SNR data.

Additionally, the major part of the signal power is concentrated on elevation angles ranging from 7 to 11 degrees (see Fig. 1). A preliminary analysis for the entire wheat growing cycle showed that, more often than not, the best elevation angle corresponding to the peak power was around 9 degrees. In this study, elevation and its change rate at 9 degree are used to represent the SNR data for all available satellite tracks (37 per day). It must be noted that this reference elevation angle is specific to the gain pattern and height of the antenna encountered in this experiment. It could present different values in other antenna configurations.

During the wheat growth cycle, preliminary tests showed that the average power spectrum could present multiple peaks together with a reduced maximum average power. This made $T_d$ unsuitable for the representation of the multipath SNR data. Under this situation the quality of the $T_d$ value was considered as poor and the data were not used. An example of $T_d$ time series is shown in Fig. 5 for GPS01 ascending tracks. Poor quality data (e.g. on 17-20 March, and 12-16 June) are indicated.

We sorted out the data acquired in two situations: (1) track data presenting more than one peak in the highest 80% percentile of the power spectrum, (2) $T_d$ value smaller by 10 seconds than the mean value of the lowest 10% of the dominant periods (e.g., $T_d < 352$ s for GPS01). This is further illustrated in Fig. 6, comparing a usable track and an unusable track. On 1 May, there is one peak in the average power spectrum (Fig. 6b), and the dominant period (456 s) obtained can be used to fit the SNR data in Fig. 6a. While on 15 June, there are two peaks in the average power spectrum as shown in Fig. 6d. Furthermore,

the maximum average power is only 0.54 which is significantly smaller than the maximum average power of 1.0 observed on 1 May 2015 (Fig. 6b). In Fig. 6c, the SNR pattern is clearly noisier, with smaller amplitudes and a less clear pattern than in Figs. 1a and Fig. 6a. This data set is unusable. A possible cause is the more inhomogeneous reflecting surface after the lodging event. The probability distribution (grey bars) of bad quality tracks among all available 37 satellite tracks is shown in Fig. 2 on a daily basis from 16 January to 15 July 2015. Most unsuitable tracks are observed during two time periods: (1)

at the beginning of spring, from 10 to 20 March, and (2) at the beginning of summer, from 12 to 26 June. The latter corresponded to lodging of vegetation, which occurred during a strong wind event and affected the reflecting surface height. The *in situ* observation of wheat height was only 39 cm on 18 June.

As shown in Sect. 2.4, vegetation effects on the SNR signal became significant after 5 March. After this date, $A_{norm}$ (black dots in Fig. 2) decreased drastically, in relation to plant growth. After 10 March, wheat height exceeded one wavelength (>

0.19 m). In addition to lower $A_{norm}$ values, an increasing number of unsuitable tracks was observed till 20 March, together with low values of the peak power (Fig. 5). During this time period, the vegetation gradually decreased the strength of the signal reflected from the soil surface and more signal was reflected by the vegetation. This triggered multiple peaks for some tracks. Such tracks were not used. When the vegetation surface completely replaced the soil surface as the dominant reflecting surface of the GNSS signal, a single peak period was observed again and its value increased in response to the rise

of the reflecting surface. For example, $T_d$ increased from 362 s (7 March) to 397 s (22 March) for GPS01 ascending tracks. Figure 5 shows that $T_d$ is not sensitive to vegetation height when vegetation height is smaller than one wavelength. Therefore, it can be concluded that this relative vegetation height (at satellite elevation of 9 degrees) retrieval technique does not work for vegetation height below one $\lambda$ (~ 0.19 m for L1) and when multiple peaks are observed in the average power spectrum.

**3.3 Vegetation height retrieval**

Figure 7 shows the retrieved vegetation height from 16 January to 15 July 2015, together with seven *in situ* vegetation height measurements and daily vegetation height simulations by ISBA. Since the original H retrievals present a marked levelling

effect, the moving average of the GPS height retrievals computed using a centred gliding window of 21 days is shown. The relative vegetation height retrievals are compared with ISBA height simulations and *in situ* height observations in Table 3. The differences between the seven *in situ* observations and the original H retrievals are -8 cm, +4 cm, -5 cm, -10 cm, -6 cm, -2 cm and -2 cm. Most of them exhibit a negative bias. In comparison with the errors between the *in situ* observations and the ISBA simulations (-5 cm, +6 cm, +10 cm, -15 cm, -3 cm, 0 cm and -61 cm), the GPS retrievals are closer to the observations on 30 March and 24 April (the third and forth *in situ* observations). On 18 June, the last height *in situ* observation before harvest is 39 cm, in relation to lodging. The GPS retrieval is very close to this value with only -2 cm error. On the other hand, the ISBA simulation on 18 June is still at 1 m with an error of -61 cm, because the wheat height was simulated without accounting for lodging. This result shows that the *in situ* GPS height retrievals are able to detect local changes in vegetation height. Figure 7 and the scores given in Table 4 show that the GPS retrievals are closer to the observed growing trend than the ISBA simulations. Additionally, the moving average height presents a much better fit to the *in situ* measurements than the raw GPS retrievals. We also compared the GPS retrievals with the ISBA model simulations. We obtained the following score values from 10 March to 11 June 2015: MAE = 8.9 cm, RMSE = 12.4 cm and $R^2$ = 0.89. Similar values were obtained for the comparison between the moving average height and ISBA simulations: MAE = 9.0 cm, RMSE = 11.6 cm and $R^2$ = 0.91.

### 3.4 Vegetation height vs. above-ground dry biomass

Figure 7 also shows that the retrieved vegetation height is related to the simulated above-ground dry biomass of the wheat (brown line). We found a linear relationship between the moving average height from GPS retrievals and the above-ground dry biomass simulated by the ISBA model from 10 March to 29 May 2015 (when the maximum vegetation height, 1 m, was measured), during the time period from tillering to flowering. The correlation coefficient between the moving average height and the above-ground dry biomass, with 81 observations, was 0.996.

A similar result was obtained using the *in situ* height and above-ground dry biomass measurements in Wigneron et al. (2002) over another wheat crop site (Triticum durum, cultivar prinqual) in spring 1993 (See Eqs. S7-S8 and Fig. S6 in the supplement).

### 4. Discussion

### 4.1. Can soil moisture be retrieved under significant vegetation effects?

Our results show that over a wheat field the vegetation gradually replaces the soil as the dominant reflecting surface when plant height becomes comparable to, or larger than one wavelength.

We tested the relationship between the multipath phase in Eq. (5) and soil moisture for the whole wheat growing cycle (Fig. 8). We found that when the vegetation effects are not significant ($A_{norm}$ > 0.78), the multipath phase correlates well (R =

0.92, N = 47, for the GPS10 descending tracks) with the *in situ* soil moisture observations (Fig. 8a). During this time period, the variation of multipath phase is about 12 degrees, for *in situ* VSM values ranging from 0.25 $m^3m^{-3}$ to 0.30 $m^3m^{-3}$. But when the vegetation effects are significant ($A_{norm}$ < 0.78), the multipath phase (without or with unwrapping, Fig. 8b and 8c) is no longer linearly related to soil moisture. For example, when vegetation height starts exceeding one wavelength, multipath phase rapidly decreases from 207 degrees to 43 degrees (between 10 and 20 March). Changes in multipath phase are disconnected from ISBA VSM simulations. This is consistent with CH16, who showed that soil moisture cannot be retrieved unless vegetation effects are corrected for.

### 4.2. Why does the locally adjusted *S* parameter differ from CH16?

In our experiment, the possible VSM retrieval duration was less than two months, in relatively wet conditions and VSM varied little: 0.25 $m^3m^{-3}$ < VSM < 0.30 $m^3m^{-3}$. This is probably not enough to represent the full yearly range of soil moisture. This might affect the representativeness of the *S* parameter (Sect. 2.4) we derived from our field observations. Furthermore, the different signal wavelength (L1 = 19.03 cm, L2 = 24.45 cm) and the different antenna gain pattern also affect the *S* parameter. Many local environment factors such as vegetation effects, precipitation, changes in soil roughness and soil composition, can perturb the GPS VSM estimates. All these factors contribute to changes in *S*, and further affect the retrieval accuracy and the sub-daily variability of VSM estimates. That is why we used a scaled soil wetness index based on the normalized multipath phase for each track, without a priori knowledge of *S* parameter. This approach also gives more accurate results.

### 4.3. Can vegetation water content be inferred from the wavelet analysis?

We found that VWC impacts the peak power but we were not able to retrieve VWC at this stage.
Figure 7 shows that the retrieved vegetation height is consistent with independent height measurements. However, vegetation height is not the only factor affecting the reflected GPS signal. Vegetation water content (VWC, in kg $m^{-2}$) may also play a role on the reflected GPS signal. *In situ* observations indicate that VWC increased together with H during the growing period, from March to mid-May. From mid-May to harvest, VWC tended to decrease but H also decreased in relation to lodging. Can this specific behavior of VWC be detected from the results of the wavelet analysis? The latter provides three quantities: the dominant period (Sect. 2.5), $A_{norm}$, and the peak power.
The amplitude ($A_{norm}$) is related to some extent to VWC (see Sect. 1). However, $A_{norm}$ is calculated assuming the relative antenna height is constant. Because the wheat height increased from 10 cm to 100 cm, the relative antenna height was reduced, and this assumption was not satisfied. This affected the estimates of the amplitude of the multipath SNR data, especially when the wheat was tall. Comparing Fig. 6a and Fig. 6c, it can be observed that the signal amplitude is larger on 1 May than that on 15 June. But $A_{norm}$ (0.15) on 1 May is even smaller than the $A_{norm}$ (0.33) on 15 June (Fig. 2). It is likely that $A_{norm}$ was underestimated on 1 May. Therefore, it is difficult to unequivocally relate $A_{norm}$ to vegetation characteristics, as

illustrated in Fig. 2. However, the drop in $A_{norm}$ observed at the beginning of June (Fig. 2) could be related to the drop in VWC.

From the wavelet analysis, we also obtained the peak power when we searched for the peak period from the average power spectrum. Peak power can represent changes in the multipath SNR strength. Figure 9 shows daily box plots of the peak power for all available satellite tracks from 16 January to 15 July 2015, together with the distribution of bad quality tracks (as in Fig. 2), and rainfall. There are two major possible causes for a sudden reduction of the strength of the SNR signal: (1) the attenuation of the signal by the rain intercepted by vegetation or in the troposphere and (2) the occurrence of more than one dominant reflecting surface at different heights, and this two causes can occur at the same time.

Three events of rapid reduction of the peak power can be observed in Fig. 9a. These events are related to larger daily standard deviation (STD) values of vegetation height retrievals (see Fig. 9b). The last event in June could be related to lodging. However, whether maximum STD is an indicator of lodging or not is unclear. It seems that these events are not related to rainfall events, and that the attenuation by intercepted water content is not a major cause of peak power drops. On the other hand, the emergence of multiple peaks and of bad quality tracks is consistent with the rapid power reduction in March and June. Multiple peaks may indicate that the reflected signal originates from surfaces at different heights. A possible cause of multiple peaks is a more heterogeneous wheat canopy density during the first stage of the growing period and after lodging. In such sparse or mixed vegetation conditions, VWC is not uniformly distributed and the soil surface may significantly contribute to the SNR. In the middle of April, there is no such effect but STD score increases (Fig. 9b). It is interesting to note that the peak power drops in Fig. 9a correspond to rapid changes in the retrieved vegetation height in Fig. 9c at multiples of $\lambda$ or $0.5\lambda$. It must be noted that absolute daily changes in $H$ (and $h$), of about 1.1 cm d$^{-1}$ are fairly uniform throughout the growing period. Since $h$ decreases when plants grow, relative changes in $h$ tend to increase. According to Eq. (4), $T$ behaves similarly. This means that the sensitivity of the retrieval method to changes in $H$ is larger at the end of the growing period. This is probably why leveling is more pronounced between mid-March and mid-April than at the end of April (see Fig. 9c). Leveling is less noticeable in May.

### 4.4. Can unwrapped multipath phase be used to retrieve vegetation height?

Our results indicate that using the dominant period to retrieve vegetation height is more relevant than using the multipath phase.

The relationship between the multipath phase (Fig. 8) in Eq. (5) and vegetation height was investigated. Because changes in relative antenna height exceeded $\lambda$ during vegetation growth, the multipath phase had to be unwrapped. When the vegetation height was smaller than $\lambda$ (before 10 March), multipath phase (around 200 degrees) presented little changes (about 12 degrees). From 21 March to 18 April, multipath phase was much smaller (around 10 degrees) and relatively stable. On the other hand, the variability increased from 19 April to 11 June (Fig. 8c), and no relationship with plant growth could be

found. It can be noted that multipath phase and dominant period are relatively stable when the vegetation height is smaller than $\lambda$. Both tend to aggregate at several value levels.

## 4.5. Can wheat phenological stages be inferred?

Figure 9 shows that the occurrence of multiple peaks together with a drop of the peak power can be used as an indicator of the start of the most active part of the growing season, and of the end of the senescence period preceding the harvest.

We applied the GDD model (see Sect. 2.6) to year 2015 and we obtained the following dates for tillering, flowering, and ripening: 12 March, 31 May, and 3 June, respectively (see Fig. S3 in the Supplement). The obtained tillering date (12 March) is close to the start date (10 March) of the multiple peaks (see Section 3.2). Tillering in wheat triggers nitrogen uptake and the accumulation of biomass (Gastal and Lemaire, 2002). This is consistent with the rapid changes in the indicators derived from the wavelet analysis: drop in $A_{norm}$ values and high rate of multiple peaks (Fig. 2), rise in the retrieved H (Fig. 7), and drop in peak power (Fig. 9). For our site, the tillering date also corresponded to the period when H reached a value of about 0.2 m. This was the case in 2015 and also in 2010 at the same site (Betbeder et al., 2016).

Flowering and ripening did not trigger abrupt changes in the GPS retrievals. However, these stages corresponded to a change in H trend. This is illustrated in Supplement Fig. S7, which shows the difference between retrieved vegetation height at a given date and retrieved vegetation height 15 days before. Flowering and ripening occur towards the end of the growing period when the vegetation height is no longer increased compared with 15 days before but slightly declines due to wheat heads tipping down (Wigneron et al., 2002). In order to confirm these findings, it could be recommended to perform GNSS-IR measurements over other wheat fields and other crops, together with phenological stage observations combined with in situ height measurements.

## 4.6 Potential future applicability and transferability of the retrieval method

*In situ* VSM observations are not widespread in France and *in situ* vegetation height observations are generally not available. Therefore, ISBA simulations are key for water resource monitoring at the country scale. It must be noted that the ISBA model is forced by the SAFRAN atmospheric analysis (Durand et al., 1993; Durand et al., 1999) and that SAFRAN is able to integrate thousands of *in situ* raingage observations. ISBA is also able to simulate vegetation characteristics such as vegetation height, leaf area index and above-ground dry biomass. However, *in situ* VSM observations are needed to validate the model simulations (e.g. Albergel et al., 2010). From this point of view, the spatial resolution of GNSS retrievals is an asset. The area sampled by GNSS retrievals is much larger than what can be achieved using individual soil moisture probes and much smaller than pixel size of satellite-derived products. Longer continuous time periods of GNSS retrievals should be envisaged to serve as independent validation data sources in statistical methods such as Triple Collocation (Dorigo et al., 2010).

We successfully assessed the surface soil moisture retrieval technique over a wheat crop field, during the start of the growing period. However, the rather narrow range of surface soil moisture values during the corresponding experiment time period limited the representativeness of the obtained retrieval accuracy. Furthermore, our dataset did not include GNSS data and *in situ* VSM measurements for periods of bare soil. Longer periods presenting a bare soil surface should be investigated in future studies. At the same time, more *in situ* vegetation measurements should be carried out.

The retrieved vegetation height was based on the dominant period of the average power spectrum. The latter was derived from GPS multipath SNR data for elevation angles between 5 and 20 degrees. We only considered the dominant period variations, without accounting for instantaneous phase changes. The accuracy of the retrieved vegetation height could probably be improved considering changes in both period and phase of the multipath SNR oscillations.

In this study, only the SNR data of L1 C/A signal is used, SNR data from different wavelength (e.g., L1 C/A, L2C and L5) should also be compared or combined to survey canopy characteristics.

A linear relationship between wheat height and dry biomass was observed during the period from wheat tillering to ripening. Retrieving dry biomass is a motivation for further research because most current satellite vegetation products focus on retrieving vegetation indexes or leaf area index. The dry biomass is directly related to the wheat yield, and retrieving wheat height could have applications in crop monitoring. In this study, only wheat is considered. Other crops should be investigated in the future.

## 5. Conclusions

GNSS SNR data were obtained using the SNR GNSS-IR technique over an intensively cultivated wheat field in southwestern France. The data were used to retrieve either soil moisture or relative vegetation height during the growing period of wheat. Vegetation growth tended to decrease the relative antenna height and broke up the constant height assumption used in soil moisture retrieval algorithms. Soil moisture could not be retrieved after wheat tillering. A new algorithm based on a wavelet analysis was implemented and used to extract the dominant period of the SNR and to retrieve vegetation height. The dominant period was derived from the peak period of the average power spectrum derived from a wavelet analysis of SNR. The method proposed by CH16 was used to retrieve soil moisture under sparse vegetation conditions, before wheat tillering. Soil moisture was retrieved on a daily basis with a precision (SDD) of 0.008 $m^3 m^{-3}$. Before tillering, only one stable peak was observed in the average power spectrum, because the soil surface was the dominant GNSS reflecting surface. During and after tillering (10-20 March), the reflected GNSS signal included contributions from both soil and vegetation. More than one peak was observed in the average power spectrum together with low values of peak power, showing that there were no clear dominant reflecting surface. Wheat growth gradually raised the reflecting surface of the GNSS signal, from the soil surface to the vegetation surface, which significantly modulated the dominant period of the multipath SNR data. In these conditions, vegetation effects could not be ignored and soil moisture could not be retrieved. The retrieved vegetation height was in good agreement with the *in situ* observations, and was

consistent with a lodging event. However, the retrieved height consisted of several levels. Using a moving average on the retrieved height permitted a better match with the *in situ* height measurements: a precision of 3.8 cm could be achieved, against 5.5 cm for the original retrievals. Furthermore, several indicators derived from the wavelet analysis could be used to detect tillering. We also found that VWC impacts the peak power but the latter cannot be used to retrieve VWC at this stage.

*Acknowledgments.* The work of Sibo Zhang and Karen Boniface was supported by the STAE (Sciences et Technologies pour l'Aéronautique et l'Espace) foundation, in the framework of the PRISM (Potentialités de la Réflectométrie GNSS In-Situ et Mobile) project. Authors would also like to thank the farmer, Mr Blanquet, for his time and the person who helped for collecting ground data.

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

**Table 1**. Soil moisture scores from 16 January to 5 March 2015.

| | GPS vs. *in situ* | GPS vs. ISBA | GPS vs. *in situ* | GPS vs. ISBA | GPS ($\varphi_{index}$) vs. *in situ* | GPS ($\varphi_{index}$) vs. ISBA | ISBA vs. *in situ* |
|---|---|---|---|---|---|---|---|
| S (m$^3$m$^{-3}$deg$^{-1}$) | 0.0148 | | 0.0033 | | - | - | - |
| N | 47 | 43 | 47 | 43 | 47 | 43 | 43 |
| MAE (m$^3$m$^{-3}$) | 0.036 | 0.034 | 0.011 | 0.018 | 0.007 | 0.009 | 0.009 |
| RMSE (m$^3$m$^{-3}$) | 0.046 | 0.041 | 0.014 | 0.022 | 0.009 | 0.012 | 0.010 |
| SDD (m$^3$m$^{-3}$) | 0.036 | 0.037 | 0.009 | 0.012 | 0.008 | 0.011 | 0.006 |
| Mean bias (m$^3$m$^{-3}$) | 0.029 | 0.019 | -0.010 | -0.018 | 0.003 | -0.005 | 0.008 |
| R$^2$ | 0.73 | 0.63 | 0.73 | 0.63 | 0.74 | 0.65 | 0.88 |

5   **Table 2**. Sub-daily variability (standard deviation, in m$^3$m$^{-3}$) of VSM estimates.

| | Minimum | Maximum | Average value |
|---|---|---|---|
| *In situ* observations | 0.000 | 0.009 | 0.002 |
| ISBA simulations | 0.000 | 0.021 | 0.005 |
| GPS retrievals with S = 0.0148 m$^3$m$^{-3}$deg$^{-1}$ | 0.012 | 0.090 | 0.036 |
| GPS retrievals with S = 0.0033 m$^3$m$^{-3}$deg$^{-1}$ | 0.003 | 0.020 | 0.008 |
| GPS retrievals from scaled soil wetness indexes | 0.005 | 0.017 | 0.009 |

Table 3. Vegetation height retrievals from GPS and simulations from ISBA, and their relative deviations for each *in situ* height observation. The phenological statuses are derived from the GDD model.

| Dates (Year 2015) | Phenological status | *in situ* height (cm) | GPS height (cm) | ISBA height (cm) | *in situ* - GPS (cm) | *in situ* - ISBA (cm) |
|---|---|---|---|---|---|---|
| 20 January | - | 10 | 18.4 | 15.4 | -8.4 | -5.4 |
| 10 March | - | 20 | 15.7 | 14.5 | 4.3 | 5.5 |
| 12 March | Tillering | - | 15.5 | 15.6 | - | - |
| 30 March | - | 35 | 40.4 | 24.6 | -5.4 | 10.4 |
| 24 April | - | 55 | 65.3 | 70.0 | -10.3 | -15.0 |
| 19 May | - | 97 | 102.9 | 100.0 | -5.9 | -3.0 |
| 29 May | - | 100 | 101.7 | 100.0 | -1.7 | 0.0 |
| 31 May | Flowering | - | 102.4 | 100.0 | - | - |
| 3 June | Ripening | - | 101.9 | 100.0 | - | - |
| 18 June | - | 39 | 40.5 | 100.0 | -1.5 | -61.0 |

10 **Table 4**. Vegetation height scores from 10 March to 11 June 2015.

| | GPS vs. *in situ* | Moving average (21 days) GPS vs. *in situ* | GPS vs. ISBA | Moving average (21 days) GPS vs. ISBA | ISBA vs. *in situ* |
|---|---|---|---|---|---|
| N | 5 | 5 | 87 | 94 | 5 |
| MAE (cm) | 5.5 | 3.7 | 8.9 | 9.0 | 6.8 |
| RMSE (cm) | 6.2 | 5.0 | 12.4 | 11.6 | 8.6 |
| SDD (cm) | 5.5 | 3.8 | 12.5 | 11.6 | 9.6 |
| Mean bias (cm) | 3.8 | 3.7 | -0.6 | -0.8 | 0.4 |
| $R^2$ | 0.98 | 0.99 | 0.89 | 0.91 | 0.95 |

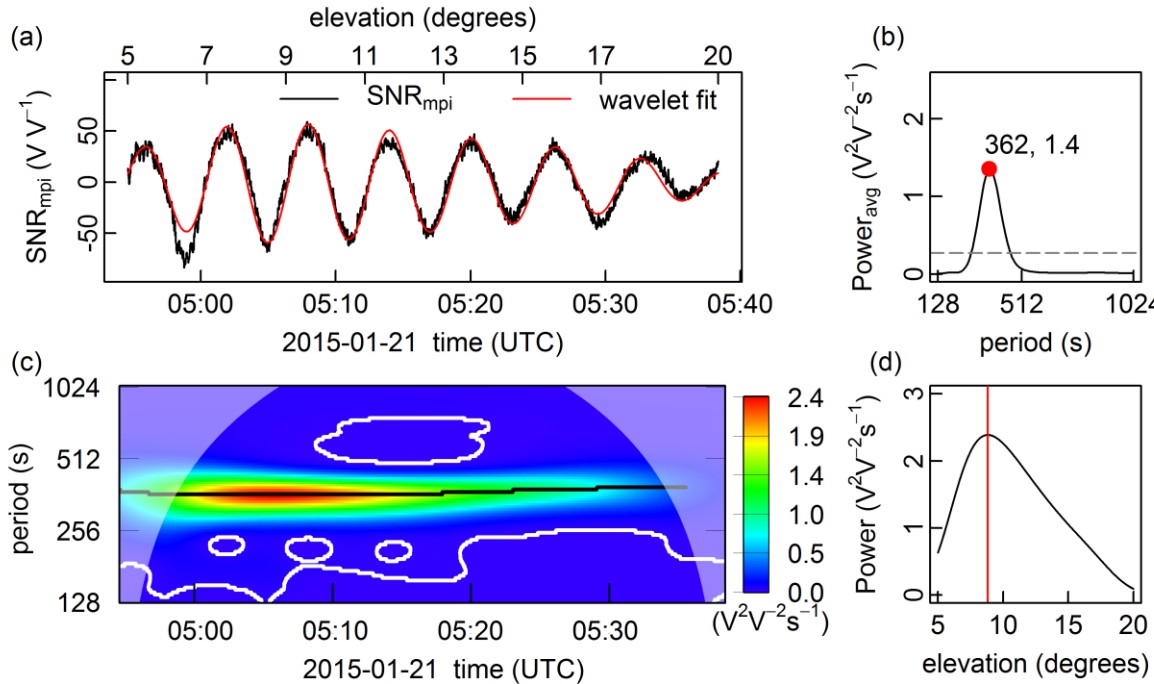

**Figure 1. Example of a usable GPS01 ascending track SNR data set from 04:50 UTC to 05:38 UTC on 21 January 2015: (a) Multipath SNR data (in V V⁻¹), (b) average power spectrum with its maximum value (red dot), and (c) power spectrum for periods from 128 to 1024 s. The red line in (a) is the reconstructed SNR data by the daughter wavelet corresponding to the peak period (362 s) indicated in (b). The power at the peak period across elevation angles (d) presents a maximum value at an elevation angle of about 9 degrees.**

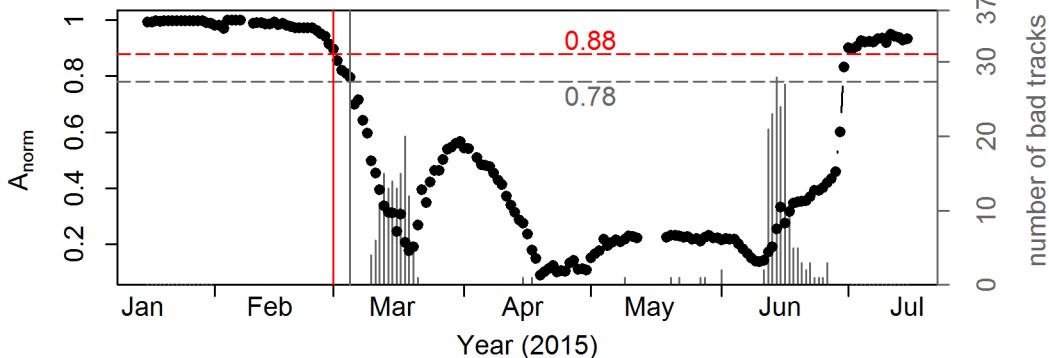

5    **Figure 2. Normalized amplitude ($A_{norm}$) time series (black dots) and probability distribution (grey bars) of low quality tracks among all available satellite tracks on a daily basis from 16 January to 15 July 2015. The empirical $A_{norm}$ threshold (0.78) is shown by the grey dashed line, and the soil moisture can be retrieved from 16 January to 5 March 2015 depending on it. Our field intuitive estimated $A_{norm}$ threshold (0.88) depending on the $A_{norm}$ in post-harvest (after 30 June) is shown by the red dashed line, and it indicates the soil moisture can be retrieved from 16 January to 1 March 2015.**

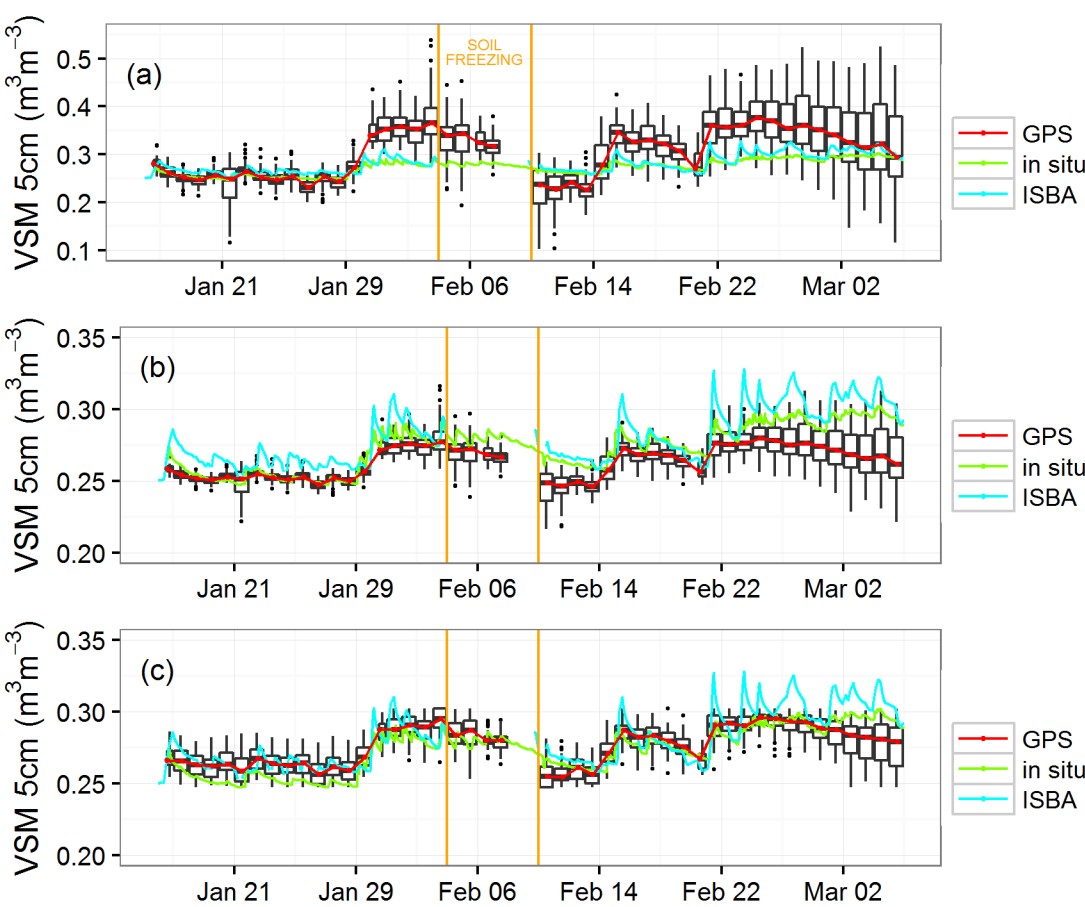

**Figure 3.** *In situ* surface volumetric soil moisture (VSM) observations at 5 cm depth (green line), ISBA simulations (blue line), median of the daily GPS retrievals (a) with the a priori slope ($S = 0.0148$ m³m⁻³degree⁻¹) (red line), (b) with a locally adjusted slope ($S = 0.0033$ m³m⁻³degree⁻¹) (red line) and (c) from scaled soil wetness index (red line), and their daily statistical distribution (black box plots) for all available satellite tracks from 16 January to 5 March 2015. Boxes: 25-75% percentiles; bars: maximum (minimum) values below (above) 1.5 IQR (Inter Quartile Range, corresponding to the 25-75% percentile interval); dots: data outside the 1.5 IQR interval. The ISBA simulations indicate soil freezing (i.e. the presence of ice in the top soil layer) from 4 to 9 February.

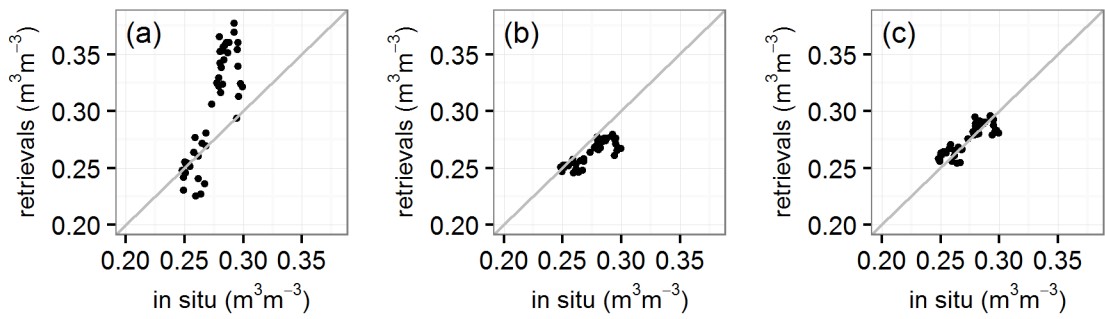

**Figure 4. VSM GPS retrievals (N = 47) versus daily mean *in situ* VSM observations (m³m⁻³) at 5 cm from 16 January to 5 March 2015, (a) with the a priori slope $S = 0.0148$ m³m⁻³degree⁻¹, VSM $= 0.0148\Delta\varphi + 0.252$, (b) with the locally adjusted slope $S = 0.0033$ m³m⁻³degree⁻¹, VSM $= 0.0033\Delta\varphi + 0.252$, and (c) from scaled soil wetness indexes, VSM $= 0.055\varphi_{index} + 0.247$. More scores can be referred from Table 1.**

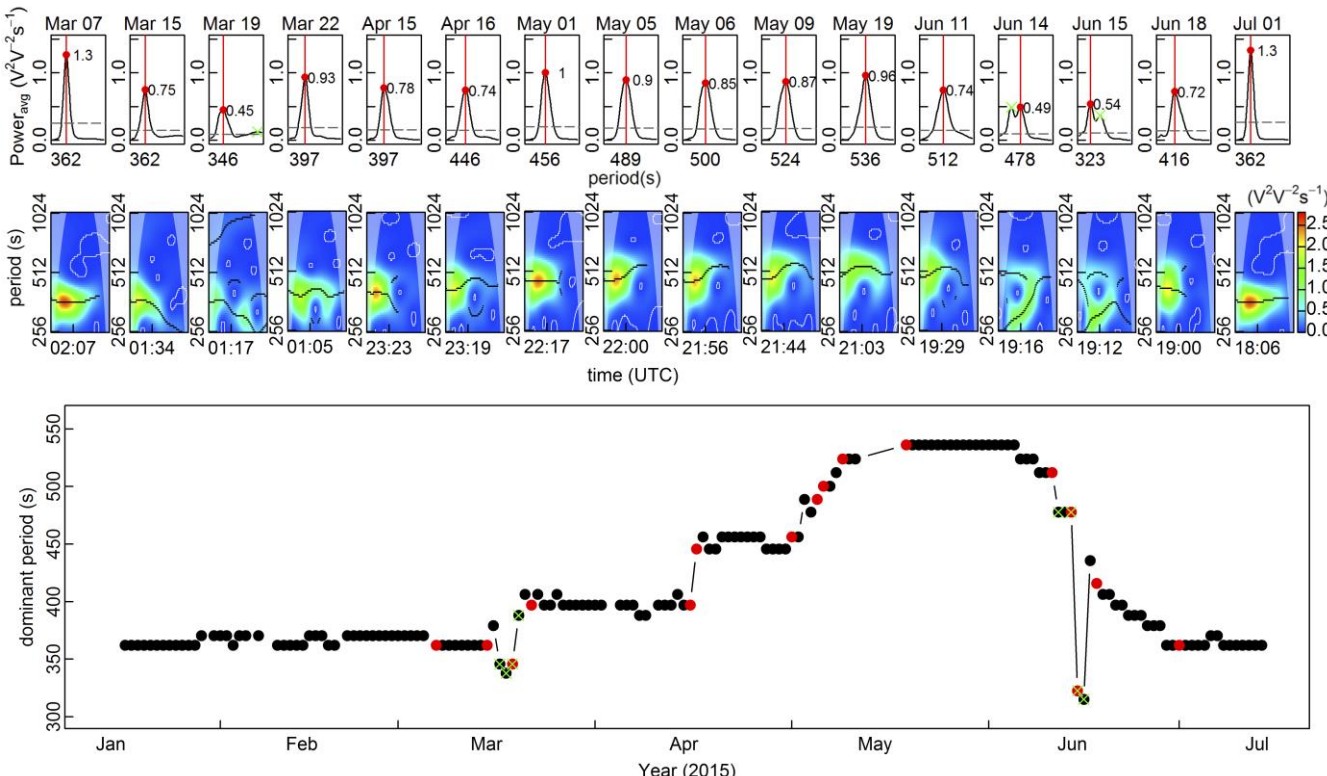

**Figure 5.** SNR dominant period ($T_d$) time series (black dots in the bottom sub-figure) derived from the GPS01 ascending tracks, with the green crosses indicate more than one peak are recognized as bad quality data, from 16 January to 15 July 2015. And (top) the average power spectrums with their maximum values (red dots), and (middle) power spectrums on the selected days (red dots in the bottom sub-figure) are also shown.

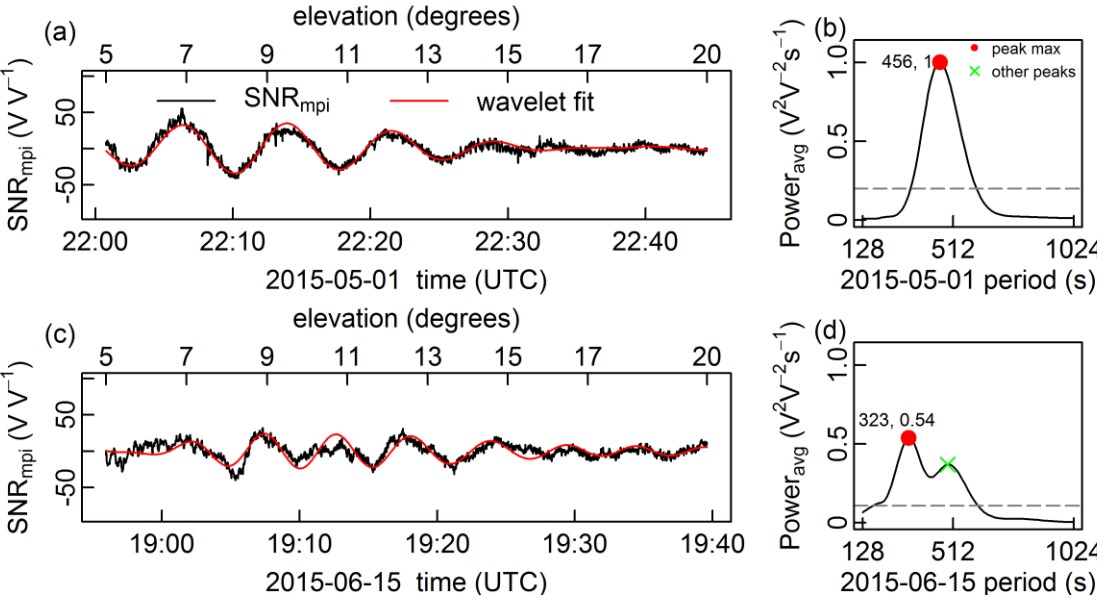

**Figure 6.** Examples of (a) usable and (c) unusable track data sets from the ascending tracks of GPS01 on1 May 2015 and 15 June 2015, respectively: (a, c) multipath SNR data, and (b, d) average power spectrums. The red lines in (a, c) are the reconstructed SNR data by the daughter wavelet corresponding to the maximum peak periods in (b, d), respectively. The green cross in (d) shows there is more than one peak in this track data, indicating bad quality, unusable data.

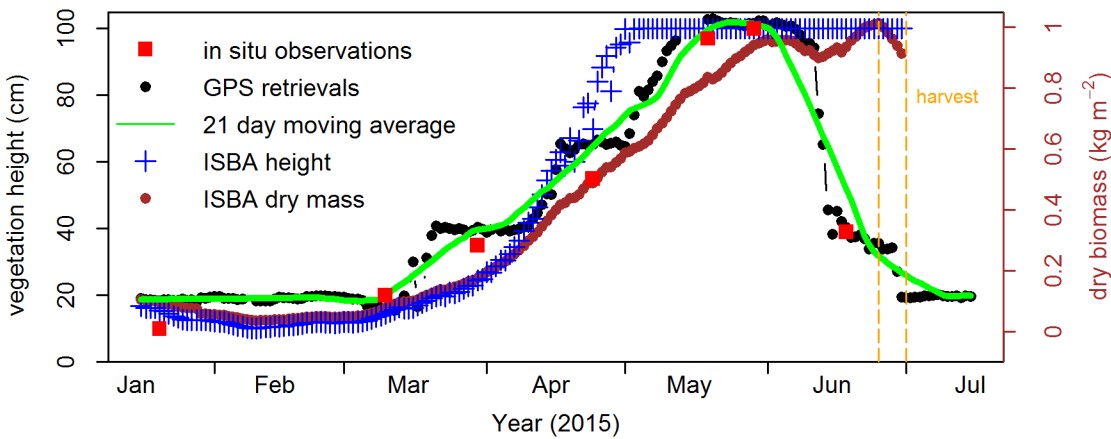

**Figure 7. Wheat canopy height from 16 January to 15 July 2015 derived from GPS SNR data (black dots), from *in situ* observations (red squares), and from ISBA simulations (blue crosses). The green line represents the moving average of the GPS retrievals, computed using a centred gliding window of 21 days. Wheat above-ground dry biomass simulated by the ISBA model is indicated by brown dots.**

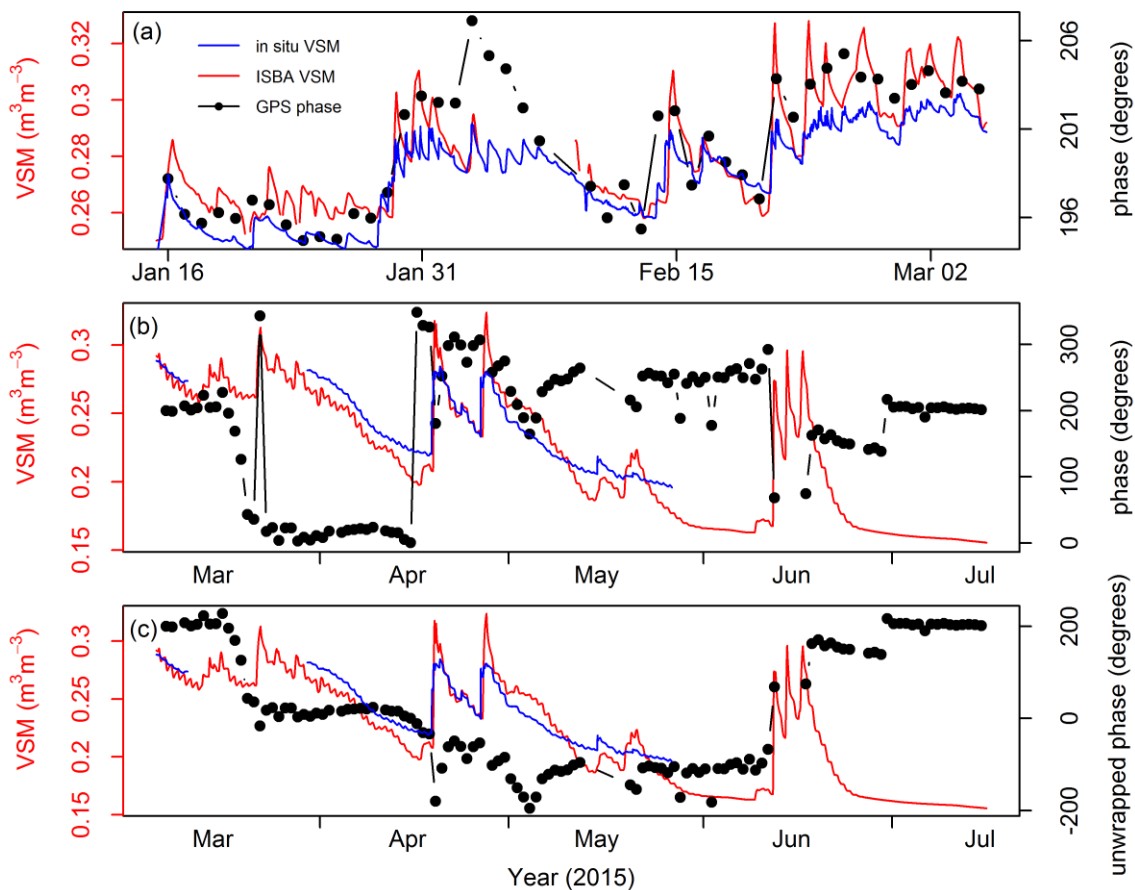

**Figure 8. Example of a track data set (descending tracks from GPS10): (a) from 16 January to 5 March, with no significant vegetation effects; (b) and (c) from 6 March to 15 July, with significant vegetation effects. In (a) and (b), multipath phases (black dots) are compared with *in situ* VSM measurements at 5 cm (blue line) and ISBA simulations (red line). In (c), unwrapped multipath phases (black dots) are used to compare with *in situ* and simulated VSM.**

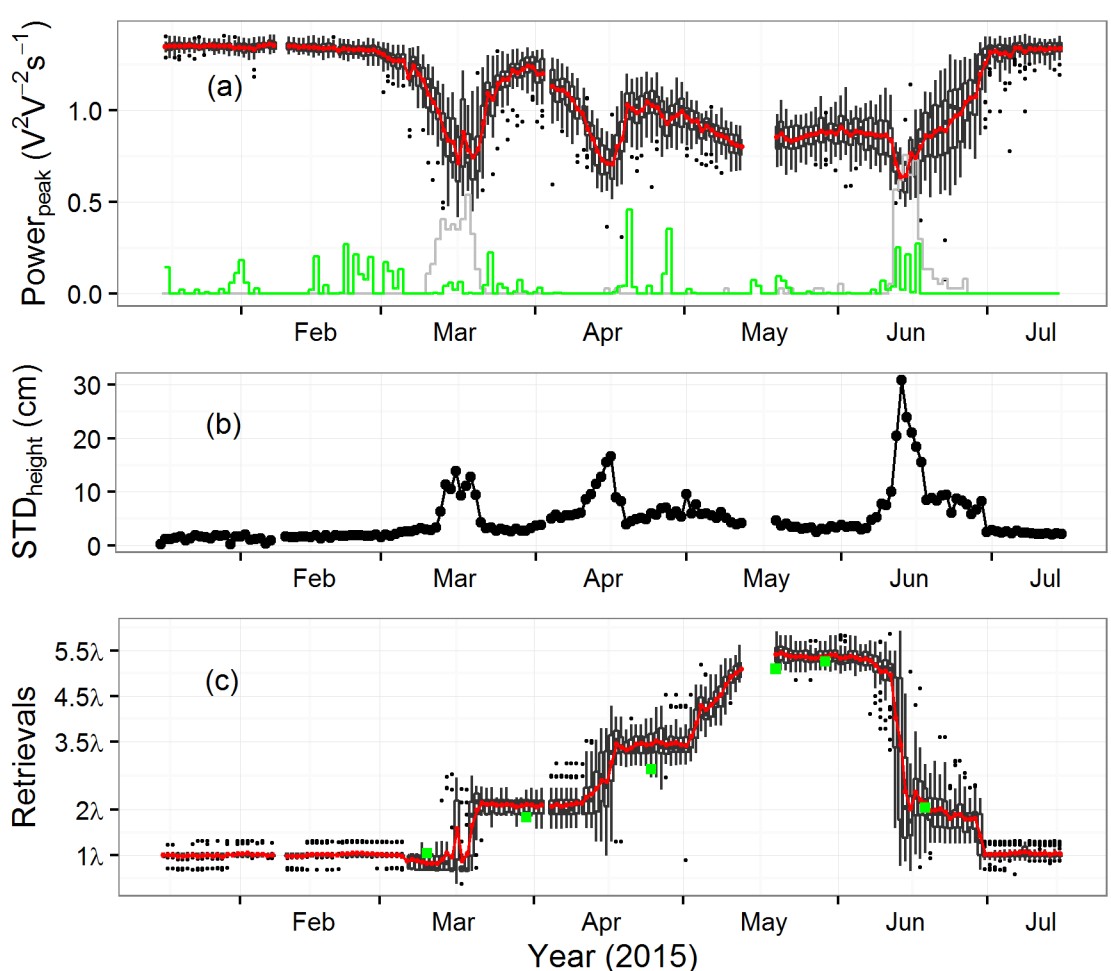

**Figure 9. The box plots of (a) the peak power from a wavelet analysis, (b) standard deviation (STD) score of the retrieved vegetation height and (c) the retrieved vegetation height (rescaled in λ units) for all available satellite tracks from 16 January to 15 July 2015. The mean value of the peak power in (a) and of the retrievals in (c) are shown by red lines. In (a), the grey line shows the statistical distribution of bad quality tracks (the number of the bad quality tracks can be obtained multiplying by 37), the green line represents the rainfall (daily precipitation in mm d$^{-1}$ can be obtained multiplying by 50). In (c), the rescaled *in situ* observations are shown by green squares.**