# Peer review of "Use of GNSS SNR data to retrieve soil moisture and vegetation variables over a wheat crop"

_Hydrology and Earth System Sciences, 2017_

## Referee Comment (RC1) · Anonymous Referee #1 · 27 Mar 2017

The study deals about soil moisture, vegetation height and phenological stages estimation by GNSS for a site in southern France and validation to in situ measurements and model simulations. The approach is sound, the manuscript well-written and adequate for the audience of HESS. Because of its high quality, just few attempts need to be made to improve the presentation of the study.

E.g., a brief discussion how much in situ (soil moisture) data is necessary to retrieve soil moisture from GNSS signal could clarify the need for adequate calibration.

During the investigation period little soil moisture variation has been recorded by in situ and GNSS sensors. The authors should discuss this low range and its relationship to the retrieval accuracy of 0.03 m3m-3. Similarly, longer time periods should be envisaged for further studies, this delivers the basis for further statistical methods such

as Triple Collocation. This would better identify the different uncertainties between the data sets. Especially with the very good results of ISBA simulations, one could question the need for (additional) GNSS measurements.

Soil moisture retrieval results could better be discussed by including recent literature and comparing to other GNSS soil moisture retrieval methods.

The authors ask the question if phenological stages can be inferred from GNSS. The outcome and visibility of the paper could be increased by giving more specific information about different stages or managements, e.g. in form of an index or threshold for wheat as an important representative for all cereals.

In total, I would recommend minor revision prior to publication.

Specific comments:

Abstract: More information about the retrieval method should be added.

P. 2, L. 10f: Refer also to the other L-band satellite SMAP.

P. 3, L. 15: introduce L2C.

P. 3, L. 26: What characterizes the dominant period?

P. 4, L. 10: Introduce PBO.

P. 5, L. 15: Start the section with explaining the aim of the calculations.

P. 6, L.10: Again, explain in one or two sentences the general concept of soil moisture retrieval before starting the details of this section.

P. 7, L. 9: A discussion about the reasons and needs for omitting a soil moisture retrieval under vegetation is necessary. Why were alternative methods not used?

P. 10, L. 3: how independent are the in situ data when some have been used for calibration? This needs to be clarified.

P. 10, L. 11ff: The reason for larger variability in GPS daily soil moisture estimates could be found in the different locations observed. During satellite overpasses the observed location moves within the larger "footprint" of the GNSS system.

P. 11, L. 12: What is the reason for using a curve smoothing procedure? What are the reasons for the levelling effect?

P. 12, L. 22ff: The authors could show the retrieval of a soil wetness index and relate it to in situ soil moisture by multiplying it to porosity (from in situ measurements or soil maps).

P. 13, L. 11: these

---

## Referee Comment (RC2) · Anonymous Referee #2 · 12 Apr 2017

General comments

This paper presents a case study applying GNSS signals, which were reflected on the ground surface (soil, vegetation surface) to derive soil moisture and vegetation height data over a wheat crop field. The GPS antenna was installed at a height of 2.51 m. Soil moisture was retrieved as long as the vegetation height was lower than ∼20 cm. However, with a further increase in plant height, it was not possible to retrieve soil moisture. Reaching a certain plant height, it was then possible to retrieve the vegetation height from the GNSS signals.

In general, the topic of this manuscript is interesting and worth to be published in HESS. The methods seem valid and transparent. However, before publishing, this manuscript has to undergo major revision as several points have to be clarified and described

/ discussed better / more clearly. The manuscript should undergo an English spell check.

The following points should be improved in general:

- Please highlight in a more prominent way what is really new and what is the outcome and applicability of this approach.

- Please introduce and explain the so called 'dominant period' in more detail.

- Please clarify that the GNSS retrieval of soil moisture and / or vegetation variables, actually only vegetation height, is only valid for different temporal stages. Especially, at the beginning it is unclear / confusing that soil moisture and vegetation height were retrieved at different time periods (before and after vegetation significant growth in March).

- If the title contains 'vegetation variables' but only 'vegetation height' is retrieved, please change this in the title and at relevant parts of the manuscript.

- The structure of the paper is not always clear – especially the chapters 'Method', 'Results' and 'Discussion' should be structured better. Some results / discussions already appear in the methods part, some points of the discussion in the results part and some methods in the discussion part.

- In some parts, the methods are explained very well, but in some parts they are presented too extensively. The manuscript should be more focused on your applied method and should be shortened as many aspects are already published in literature and don't have to be repeated in this manuscript.

- Is it necessary to retrieve soil moisture before retrieving vegetation height? Please comment on this.

- Regarding the statistics, 7 or even only 5 (during the period you used to demonstrate vegetation height) in situ vegetation height samples are actually too low. Please

comment at least that during further studies more in situ data should be carried out.

- It is questionable if all information given in the supplement is needed. On the other hand, some figures (see specific comments below) would also be valuable within the manuscript itself and should be presented there.

Specific comments

Page 1 – Title

Please clarify that the retrieval of soil moisture and vegetation variables are actually only valid for different temporal stages (before and after vegetation significant growth in March). Moreover, it would be valuable to include that you use reflected GNSS signals in your approach as also other GNSS approaches exist on this topic.

Title suggestion: 'Use of reflected GNSS SNR data to retrieve either soil moisture or vegetation height, depending on the vegetation phase of a wheat crop field,'

Page 1 – Abstract

General: The absolute length of the abstract seems fine, however, the information given here should be compressed or information should be combined more functionally. Additionally, it should be added why this approach is generally useful (1 sentence) and what is missing so far regarding the state of art (1 sentence)

p.1, l.15: '...numerical simulations of biomass...'

p.1, l. 18: describe in few words the 'dominant period'

p.1, l. 18: '...SNR data, whereas changes in...'

p.1, l. 20: '...smaller than one wavelength ($\sim$19 cm).' This should also be changed in the entire manuscript.

p.1, l. 22: dry biomass?

Page 1-3: 1. Introduction

General: The introduction is quite good, but it should be written more comprehensively, especially the parts where you describe already published techniques. However, the first part (p.1, l.27-p.2, l.2) where you introduce the necessity of this approach and the recent lack to monitor land surface variables at a local scale should be extended! Moreover, it should be written more clearly why GNSS reflectometry could be a solution.

p.2, l.7: The frequency of GPS L1-band is 1.57542 GHz. Please write 1.6 GHz instead of 1.5 GHz.

p.2, l.10: 'These properties have e.g. been. . .'

p.2, l.10-15: As you generally mention L-band active and passive remote sensing techniques, also other GNSS methods (besides reflectometry) aiming to derive soil moisture or vegetation parameters should be mentioned (e.g. GNSS methods using signal attenuation).

p.2, l.17: please specify, how these two antennas are mounted?

p.2, l.26: 'They are surrounded by sparse vegetation and are therefore not useful for vegetation studies.'

p.3, l.31/32: Better write 'lower and taller vegetation' as you are measuring the vegetation height and not their density.

Page 4-5: 2. Data

General: Actually this section already belongs to the 'Method' section.

p.4, l.4: Fig. S1: This figure is not really valuable to show where the test field is situated (present either a picture of the GNSS antenna in the field or a map where the field is situated)

p.4, l.14: '. . ., four GPS satellites of in total 32. . .

p.4, l.18: refer to relevant figure
p.4, l.1-2 and l.29: avoid repetitions

p.4, l.1ff: add information on the soil type and texture; moreover, the row spacing of the wheat crop would be interesting.

p.4, l.30/31: which satellite observations are meant? GNSS satellites or EO satellites?

p.5, l.2: 'soil moisture and vegetation height. . .'

p.5, l.5: Which soil moisture instruments did you use as reference, e.g. frequency domain probes?

p.5, l.8: add the vegetation height at the end of the season as well. Moreover, for each reference sample the measured height and the phonological status of the wheat crop would be interesting (e.g. listed in a table).

p.20, Fig. 1: Figure sub-captions (a-d) are not well structured; a legend in plot a) would be helpful (red and black line); please insert units if there are in y-axis of plot b) and plot d) and in the legend of plot c) (otherwise write []); the mentioned 128 to 1024 s are not shown in plot c –please mark or show tem additionally in a second x-axis; for more clarity in the manuscript, refer to Fig. 1a, 1b, 1c, 1d, not only to Fig. 1.

Page 5-9: Methods

General: This chapter should be written more comprehensively and precisely, especially the parts of already known methods.

p.6, l.8ff: How many soil moisture and vegetation height results per day did you get out of the 37 available satellite tracks? As of Table 1 and Table 3 it seems that you got 1 results for each day. Please clarify (short) already at this point the temporal resolution and the daily composition of your retrieved results.

p.7, l.3: Is there any S-value specific for L1 already available in literature? Or is the mentioned and an adjusted S-value for the first time applied for L1-band signals? Then this should be introduced more prominently in the manuscript!

[Figure]

p.7, l.5: It seems more logical to introduce the adjusted S-value in this chapter instead in the 'Results' chapter.

p.7, l.6ff: Perhaps it also makes sense to introduce your experimental A_norm threshold of 0.88 within this chapter. Moreover, Fig 2 should be combined with / replaced by Fig. S7.

p.7,12/13: are GNSS data available for periods of bare soil (e.g. before the wheat crops reached a vegetation height of 10 cm before January 16th) – this would be valuable to improve the final soil moisture estimate.

p.7, l.21: 'see the Supplement' – which figure or part do you mean?

p. 8, l.9ff: 'One possible reason. . .' This part fits better to the 'Discussion' part.

p.9, l.8ff: In my opinion, the 'scores' don't have to be introduced with equations.

p.21, Fig. 2: Should/could be combined with Fig. S7.

Figure S3 and S4: Especially Fig. S4 is interesting. It should be demonstrated within the manuscript as it shows at which stages it is difficult to retrieve the results according to the dominant period.

Page 10-11: Results

p.10, l.3ff: Please insert also the mean soil moisture values of each method (for the entire observation period).

p.10, l. 3-24 and p.23, Fig. 4: Is it generally possible to compare these three methods one by one? The model simulates the first 10 cm; the reference measurements record at a soil depth of 5 cm and the GPS technique observes the soil surface. Perhaps the results with a S-value of S=0.0148 are even more realistic!? Please state on this. The GPS retrieval seems to be slightly too low in this plot using a S-value of 0.0033; especially after soil freezing and at the end of the soil moisture retrieval period the correlation between GPS retrievals and observations / reference measurements is

weaker.

p.10, l. 14: 'a priori'

p.11, l.6: delete '(not shown)'

p.11, l.10ff: Please insert also the vegetation height determined either by GNSS or manually for each date, instead only listing the deviations.

p.11, l.12: why do you use a 21 gliding window approach? Is this really necessary? Perhaps the vegetation height levels in Figure 6 make sense (e.g. due to meteorological events and plant growth spurts)?

p.11, l.26: Please state more on the overall possibility to compare dry biomass and vegetation height. Is this really possible? Are there some references available? Please state on this more detailed.

p.22/23, Fig. 3/4: for better comparability, in both Figures the y-axis should have the same scale; They could also be combined in one figure with sub-figures a) and b).

p.24, Fig. 5: How many dots are shown in this plot (N=47?)? Please add this information in the figure capture.

p.25, Fig 6: You don't have to repeat the legend in the figure column.

Page 12-14: Discussion

General: The idea of asking questions is good. Please also insert a discussion section / further question on the potential future applicability and transferability (e.g. to other soils, other vegetation types, other GNSS signals etc.). What could be improved. . .

p.12, l.3ff: As important findings (regarding the discussion) are shown in Fig. S6, this figure should also be shown in the manuscript (not only the supplement). Moreover, this issue should be discussed in more detail.

p.12, l.9: Why did you increase this threshold exactly to the value 0.88? Is there any

reason for this value?

p.12, l.26: Re-formulate your question: 'Can other vegetation characteristic besides vegetation height be inferred from the wavelet analysis?'. Or formulate two questions: 'Can vegetation height be inferred from...?' and 'Is it possible to additionally retrieve other vegetation characteristics from...?'

p.12, l.27ff: The idea that you potentially also would like to retrieve the plant water content (or even other vegetation characteristics) should already be introduced earlier in the manuscript. Then an answer to this question would make more sense in the 'Discussion' part. Do you have reference data that show a decrease in plant water content?

p.13, l.22: What do you mean with STD?

p.13, l.17: The rainfall/meteorological and logging events could additionally be shown in the figure, e.g., as a subplot.

p.14, l.2-8: This actually belongs to the 'Method' chapter. It is a further method to compare your retrievals to a reference.

Page 14-15: Conclusions

General: Give also an outlook on potential applicability of this technique.

p.14, l.19: Please specify – is this a new algorithm you developed or do you mean at this point the algorithm of CH15 and others you applied for the wheat crop test field?

p.15, l.2: L5 is introduced here for the first time. It could be mentioned already earlier (e.g., in the 'Discussion').

Supplement

S p.1, Fig. S1: see comment above.

S p.2, Fig. S2: Applying the same time scale in the x-axis of the two plots would be

better for comparability or it would even be more helpful if both plots would be combined in one figure (e.g. with two different colours).

S p.4, Fig. S3: a legend would be useful; it would be logical for comparison to combine Fig. S3 and Fig. S8

S p.5, Fig. S4: see comment above; insert a legend and units if needed.

S p.6, Fig. S5: see comment above; how many dots are shown in this plot (N=47?)? Please add this information in the figure capture.

S p.7, Fig. S6: please add the black dots also to the legend; regarding the blue line / dots: use either dots or lines for all of the three plots.

S p.8, Fig. S7: see comment above.

S p.9, Fig. S8: see comment above.

S p.9, Fig. S9: This information could visually be combined with Fig. 2 / Fig S8.

S p. 11: Duveiller et al. 2011 should also be added to the references in the manuscript.

S p. 12: Please clarify the figure capture. Was is actually meant with '...the value retrieved 15 days before, ...'? The dates of flowering and ripening should also occur in the figure or at least in the figure capture.

---

## Author Comment (AC2) · 29 May 2017

Zhang et al.: Use of GNSS SNR data to retrieve soil moisture and vegetation variables over a wheat crop, Hydrol. Earth Syst. Sci. Discuss., doi:10.5194/hess-2017-152, 2017.

**RESPONSE TO REVIEWER #2**

The authors thank anonymous reviewer 2 for his/her review of the manuscript and for the fruitful comments.

**2.1 [General comments**

This paper presents a case study applying GNSS signals, which were reflected on the ground surface (soil, vegetation surface) to derive soil moisture and vegetation height data over a wheat crop field. The GPS antenna was installed at a height of 2.51 m. Soil moisture was retrieved as long as the vegetation height was lower than ~20 cm. However, with a further increase in plant height, it was not possible to retrieve soil moisture. Reaching a certain plant height, it was then possible to retrieve the vegetation height from the GNSS signals.

In general, the topic of this manuscript is interesting and worth to be published in HESS. The methods seem valid and transparent. However, before publishing, this manuscript has to undergo major revision as several points have to be clarified and described / discussed better / more clearly. The manuscript should undergo an English spell check. The following points should be improved in general:

- Please highlight in a more prominent way what is really new and what is the outcome and applicability of this approach.]

**Response 2.1:**

Yes. We will revise the abstract and conclutions to highlight the new results presented in this study.

In particular, the following information will be given:

GNSS SNR data were obtained using the GNSS-IR technique over an intensively cultivated wheat field in southwestern France. The data were used to retrieve either soil moisture or vegetation height during the growing period of wheat. Vegetation growth tended to decrease the relative antenna height and broke up the constant height assumption used in soil moisture retrieval algorithms. Soil moisture could not be retrieved after wheat tillering. A new algorithm based on a wavelet analysis was implemented and used to extract the dominant period of the SNR and further to retrieve vegetation height.

Should a revised version of this paper be accepted in HESS, a copy editing work will be performed.

**2.2 [- Please introduce and explain the so called 'dominant period' in more detail.]**

**Response 2.2:**

A vegetation height retrieval algorithm is proposed using the dominant SNR period, which is the peak period in the average power spectrum derived from a wavelet analysis of SNR. We will clarify this in the revised manuscript.

2.3 [- Please clarify that the GNSS retrieval of soil moisture and / or vegetation variables, actually only vegetation height, is only valid for different temporal stages. Especially, at the beginning it is unclear / confusing that soil moisture and vegetation height were retrieved at different time periods (before and after vegetation significant growth in March)].

**Response 2.3:**

Yes. We will replace "vegetation variables" or "vegetation characteristics" by "vegetation height". We will mention in the Abstract that soil moisture and vegetation height were retrieved at different time periods (before and after vegetation significant growth in March).

2.4 [- If the title contains 'vegetation variables' but only 'vegetation height' is retrieved, please change this in the title and at relevant parts of the manuscript.]

**Response 2.4:**

Yes. We will modify the title as 'Use of reflected GNSS SNR data to retrieve either soil moisture or vegetation height over a wheat crop'. We will replace "vegetation variables" or "vegetation characteristics" by "vegetation height" in the entire manuscript.

2.5 [- The structure of the paper is not always clear – especially the chapters 'Method', 'Results' and 'Discussion' should be structured better. Some results / discussions already appear in the methods part, some points of the discussion in the results part and some methods in the discussion part.]

**Response 2.5:**

Yes. We will revise the manuscript accordingly. In particular, description of Fig. 1 will be moved from Sections 2.1 and 3.2 to Section 4. Description of Fig. 2 will be moved from Sections 3.1 and 3.2 to Section 4. Description of Figs. S3 and S4 will be moved from Section 3.2 to Section 4.

2.6 [- In some parts, the methods are explained very well, but in some parts they are presented too extensively. The manuscript should be more focused on your applied method and should be shortened as many aspects are already published in literature and don't have to be repeated in this manuscript.]

**Response 2.6:**

Yes. We will try to improve the focus of Section 3. Note however that all readers of HESS are not familiar with GNSS reflectometry and that Eqs. 1-6 need to be presented.

2.7 [- Is it necessary to retrieve soil moisture before retrieving vegetation height? Please comment on this.]

**Response 2.7:**

No. It is not necessary to retrieve soil moisture before retrieving vegetation height. This will be made clear in the revised manuscript.

2.8 [- Regarding the statistics, 7 or even only 5 (during the period you used to demonstrate vegetation height) in situ vegetation height samples are actually too low. Please comment at least that during further studies more in situ data should be carried out.]

**Response 2.8:**

Yes. The in situ vegetation height samples are few, but it must be noted that GNSS height retrievals are totally independent from the in situ measurements. We will make clear that in further studies, more in situ data enabling the characterization of vegetation would be needed.

2.9 [- It is questionable if all information given in the supplement is needed. On the other hand, some figures (see specific comments below) would also be valuable within the manuscript itself and should be presented there.]

**Response 2.9:**

Yes. We will adjust this in the revised manuscript.

**2.10 [Specific comments**

Page 1 – Title

Please clarify that the retrieval of soil moisture and vegetation variables are actually only valid for different temporal stages (before and after vegetation significant growth in March). Moreover, it would be valuable to include that you use reflected GNSS signals in your approach as also other GNSS approaches exist on this topic.

Title suggestion: 'Use of reflected GNSS SNR data to retrieve either soil moisture or vegetation height, depending on the vegetation phase of a wheat crop field,']

**Response 2.10:**

Yes, we will change the title accordingly: 'Use of reflected GNSS SNR data to retrieve either soil moisture or vegetation height over a wheat crop'

2.11 [Page 1 – Abstract

General: The absolute length of the abstract seems fine, however, the information given here should be compressed or information should be combined more functionally. Additionally, it should be added why this approach is generally useful (1 sentence) and what is missing so far regarding the state of art (1 sentence)]

**Response 2.11:**

Yes. Surface soil moisture can be retrieved based on the linear relationship between in situ soil moisture observations and SNR phases estimated by the Least Square Estimation method, assuming the relative antenna height is constant. However, it is found in this study that the vegetation growth breaks up the constant relative antenna height assumption, and modulates the SNR period. A vegetation height retrieval algorithm is proposed using the SNR dominant period, which is the peak period in the average power spectrum derived from a wavelet analysis of SNR.

We will rephrase the abstract accordingly.

**2.12 [p.1, 1.15: '...numerical simulations of biomass...']**

**Response 2.12:**

Yes. The sentence will be modified as:

"The retrievals are compared with two independent reference datasets: *in situ* observations of soil moisture and vegetation height, and numerical simulations of soil moisture, vegetation height and above-ground dry biomass from the ISBA (Interactions between Soil, Biosphere and Atmosphere) land surface model."

**2.13 [p.1, l. 18: describe in few words the 'dominant period']**

**Response 2.13:**

A vegetation height retrieval algorithm is proposed using the dominant SNR period, which is the peak period in the average power spectrum derived from a wavelet analysis of SNR. We will clarify this in the revised manuscript.

2.14 [p.1, l. 18: '...SNR data, whereas changes in...']

**Response 2.14:**

Yes. The sentence will be rephrased accordingly.

2.15 [p.1, l. 20: '...smaller than one wavelength (~19 cm).' This should also be changed in the entire manuscript.]

**Response 2.15:**

Yes. We will correct it.

"Surface volumetric soil moisture can be estimated ( $R^2 = 0.73$ , RMSE = 0.014 m3m-3) when the wheat is smaller than one wavelength (~ 19 cm)."

**2.16 [p.1, l. 22: dry biomass?]**

**Response 2.16:**

Yes. We will correct this.

**2.17 [Page 1-3: 1. Introduction**

General: The introduction is quite good, but it should be written more comprehensively, especially the parts where you describe already published techniques. However, the first part (p.1, 1.27-p.2, 1.2) where you introduce the necessity of this approach and the recent lack to monitor land surface variables at a local scale should be extended! Moreover, it should be written more clearly why GNSS reflectometry could be a solution.]

**Response 2.17:**

In situ VSM observations are not widespread in France and in situ vegetation height observations are generally not available. Therefore, ISBA (Interactions between Soil, Biosphere and Atmosphere) simulations are key for water resource monitoring at the country scale. It must be noted that the ISBA model is forced by the SAFRAN atmospheric analysis and that SAFRAN is able to integrate thousands of in situ raingage observations. ISBA is also able to simulate vegetation characteristics such as vegetation height, leaf area index, and above-ground dry biomass. However, in situ VSM observations are needed to validate land surface models and/or satellite-derived products (e.g. Albergel et al., 2010). From this point of view, the spatial resolution of GNSS retrievals is an asset. The area sampled by GNSS retrievals is much larger than what can be achieved using individual soil moisture probes and much smaller than pixel size of satellite-derived products. Longer time periods of GNSS retrievals should be envisaged to serve as independent validation data sources in statistical methods such as Triple Collocation (Dorigo et al., 2010).

We will incorporate this material in the revised manuscript.

**References:**

Albergel, C., J.-C. Calvet, P. de Rosnay, G. Balsamo, W. Wagner, S. Hasenauer, V. Naemi, E. Martin, E. Bazile, F. Bouyssel, J.-F. Mahfouf, "Cross-evaluation of modelled and remotely sensed surface soil moisture with in situ data in southwestern France", Hydrol. Earth Syst. Sci., 14, 2177–2191, 2010b.

Dorigo, W. A., Scipal, K., Parinussa, R. M., Liu, Y. Y., Wagner, W., de Jeu, R. A. M., and Naeimi, V.: Error characterisation of global active and passive microwave soil moisture datasets, Hydrol. Earth Syst. Sci., 14, 2605–2616, doi:10.5194/hess-14-2605-2010, 2010.

2.18 [p.2, 1.7: The frequency of GPS L1-band is 1.57542 GHz. Please write 1.6 GHz instead of 1.5 GHz.]

**Response 2.18:**

Yes. We will correct it.

"GNSS satellites operate at the L-band microwave frequency domain (between 1.2 GHz and 1.6 GHz). "  $\,$

2.19 [p.2, 1.10: 'These properties have e.g. been...']

Response 2.19:

Yes. We will correct it.

2.20 [p.2, 1.10-15: As you generally mention L-band active and passive remote sensing techniques, also other GNSS methods (besides reflectometry) aiming to derive soil moisture or vegetation parameters should be mentioned (e.g. GNSS methods using signal attenuation).]

**Response 2.20:**

Yes. We will cite a reference using GNSS signal strength attenuation.

Larson et al. (2008) showed that SNR data obtained from existing networks with single ground-based geodetic GNSS-IR antenna can be used to infer soil moisture. Other GNSS methods (besides reflectometry) can be used. For example, Koch et al. (2016) used three geodetic GNSS antennas (one was installed above the soil, the other two were buried at a depth of 10 cm), to measure the GNSS signal strength attenuation and to retrieve soil moisture over bare soil.

**References:**

Koch, F., Schlenz, F., Prasch, M., Appel, F., Ruf, T. and Mauser, W.: Soil moisture retrieval based on GPS signal strength attenuation, Water, 8(7), 276, 2016.

2.21 [p.2, 1.17: please specify, how these two antennas are mounted?]

**Response 2.21:**

Yes. We will clarify it in the revised manuscript.

"(1) waveform acquisition with a specific receiver using two antennas (one zenith-oriented antenna and one surface-oriented antenna), called GNSS reflectometry (GNSS-R) (Zavarotny et al., 2014) or (2) GNSS signal strength, Signal-to-Noise Ratio (SNR), acquisition with classical geodetic receiver using one antenna, called GNSS interferometric reflectometry (GNSS-IR) technique (Larson, 2016). "

2.22 [p.2, 1.26: 'They are surrounded by sparse vegetation and are therefore not useful for vegetation studies.']

**Response 2.22:**

Yes. The sentence will be modified accordingly.

2.23 [p.3, 1.31/32: Better write 'lower and taller vegetation' as you are measuring the vegetation height and not their density.]

Response 2.23:

Yes. This sentence will be rephrased accordingly.

**2.24 [Page 4-5: 2. Data**

General: Actually this section already belongs to the 'Method' section. p.4, 1.4: Fig. S1: This figure is not really valuable to show where the test field is situated (present either a picture of the GNSS antenna in the field or a map where the field is situated)]

**Response 2.24:**

Yes. We will reorganize Sections 2 and 3 in a single "Materials and methods" Section. We will present a picture of the GNSS antenna in the field (see Fig. R2.1).

---

## Author Response (AR2)

**"Use of reflected GNSS SNR data to retrieve either soil moisture or vegetation height over a wheat crop"**
**by Sibo Zhang et al.**

**Cover letter to the editor**

28 August 2017

Dear Dr. Alberto Guadagnini,

In response to your comment ("*I do think that the Authors have taken advantage of the constructive comments emerged during the review process and the manuscript has considerably improved. I see no particular reasons to further delay acceptance. I do recommend a thorough check of the use of the English language by the Authors, though, which is some cases appears to be awkward*"), we checked the English and the revised text can be found enclosed.

Yours sincerely,

Jean-Christophe Calvet, Sibo Zhang.

[revised manuscript text omitted]

**by Sibo Zhang et al.**

**Cover letter to the editor**

8 August 2017

Dear Dr. Alberto Guadagnini,

The authors' response to the comments of the two anonymous referees has been published on the HESS web site. The list of all relevant changes made in the manuscript can be found in the enclosed document.
All changes relative to the published HESS paper are detailed in the pdf of the new manuscript. They include all the response elements given by the authors in response to the reviewers' comments (orange and blue for Reviewer 1 and 2, respectively). Other changes in the text are in red.

The title was changed in response to the comments of Reviewer 2.
Some parts of the paper were re-structured at the request of the reviewers. Former Sections 2 and 3 were merged in a new "Material and methods" Section. Former Section 3.3 was moved to the Supplement. The result Section was reorganised.
All the Figures were revised. In particular, former Figures 3 and 4 were merged. Former Figures S3, S4, S6, S9 and S10 are now embedded into the main text.

References

18 additional references were added (Albergel et al. 2010, Chan et al. 2016, Chew et al. 1997, Darrozes et al. 1997, Dorigo et al. 2010, Durand et al. 1993, 1999, Duveiller et al. 2011, Escalera et al;, 1995, Gaillot et al. 1999, Grinsted et al. 2004, Hagelberg et al. 1995, Koch et al. 2016, Labat 2005, Ouillon et al. 1995, Torrence and Compo 1998, Wigneron et al. 2002).

Yours sincerely,

Jean-Christophe Calvet, Sibo Zhang.

**Zhang et al.: Use of GNSS SNR data to retrieve soil moisture and vegetation variables over a wheat crop, Hydrol. Earth Syst. Sci. Discuss., doi:10.5194/hess-2017-152, 2017.**

**RESPONSE TO REVIEWER #1**

The authors thank anonymous reviewer 1 for his/her review of the manuscript and for the fruitful comments.

**The revised text is shown in orange in the enclosed version of the manuscript.**

1.1 [The study deals about soil moisture, vegetation height and phenological stages estimation by GNSS for a site in southern France and validation to in situ measurements and model simulations. The approach is sound, the manuscript well-written and adequate for the audience of HESS. Because of its high quality, just few attempts need to be made to improve the presentation of the study. E.g., a brief discussion how much in situ (soil moisture) data is necessary to retrieve soil moisture from GNSS signal could clarify the need for adequate calibration.]

**Response 1.1:**

Retrieving absolute VSM values in $m^3 m^{-3}$ is possible after a calibration phase. The minimum VSM has to be derived from the *in situ* observations during the experimental time period in order to determine the $VSM_{resid}$ term in Eq. (6). Moreover, a locally adjusted value of the *S* parameter is needed. The retrieval of the *S* parameter requires at least one or two months of VSM *in situ* observations because soil moisture conditions ranging from dry to wet need to be sampled. However, if a scaled soil wetness index is used instead of soil moisture (see Response 1.17), no *in situ* VSM observations are needed. This aspect was clarified in the revised manuscript (P. 8, L. 3-7).

1.2 [During the investigation period little soil moisture variation has been recorded by in situ and GNSS sensors. The authors should discuss this low range and its relationship to the retrieval accuracy of 0.03 $m^3 m^{-3}$.]

**Response 1.2:**

Yes, a short period of time is considered in this study. Vey et al. (2015) used the method from Chew et al. using field observations over a long period of time (2008-2014) for a site presenting a high percentage of bare soil. They obtained the following scores for GPS VSM retrievals: $R^2 = 0.8$, RMSE = 0.05 $m^3 m^{-3}$. We successfully assessed this method for a wheat crop field. But the little soil moisture variation in the experiment time period limited the representativeness of the retrieval accuracy. Longer time periods should be investigated in further studies. We clarified this in the revised manuscript (P. 17, L. 9-11).

1.3 [Similarly, longer time periods should be envisaged for further studies, this delivers the basis for further statistical methods such as Triple Collocation. This would better identify the different uncertainties between the data sets. Especially with the very good results of ISBA simulations, one could question the need for (additional) GNSS measurements.]

**Response 1.3:**

Yes. In situ VSM observations are not widespread in France and the ISBA simulations are key for water resource monitoring at the country scale. It must be noted that the ISBA model is forced by the SAFRAN atmospheric analysis and that SAFRAN is able to integrate thousands of in situ raingage observations over France. However, in situ VSM observations are needed to validate land surface models and/or satellite-derived products (e.g. Albergel et al., 2010). From this point of view, the spatial resolution of GNSS retrievals is an asset. The area sampled by GNSS retrievals is much larger than what can be achieved using individual soil moisture probes and much smaller than pixel size of satellite-derived products. Longer time periods of GNSS retrievals should be envisaged to serve as independent validation data sources in statistical methods such as Triple Collocation (Dorigo et al., 2010). This aspect was clarified in the revised manuscript (P. 16, L. 29-30; P. 17, L. 1-8).

References:

Albergel, C., J.-C. Calvet, P. de Rosnay, G. Balsamo, W. Wagner, S. Hasenauer, V. Naemi, E. Martin, E. Bazile, F. Bouyssel, J.-F. Mahfouf, "Cross-evaluation of modelled and remotely sensed surface soil moisture with in situ data in southwestern France", Hydrol. Earth Syst. Sci., 14, 2177–2191, 2010b.

Dorigo, W. A., Scipal, K., Parinussa, R. M., Liu, Y. Y., Wagner, W., de Jeu, R. A. M., and Naeimi, V.: Error characterisation of global active and passive microwave soil moisture datasets, Hydrol. Earth Syst. Sci., 14, 2605–2616, doi:10.5194/hess-14-2605-2010, 2010.

1.4 [Soil moisture retrieval results could better be discussed by including recent literature and comparing to other GNSS soil moisture retrieval methods.]

**Response 1.4:**

The method from Chew et al. is the latest proposed method, as far as we know. We will further increase the accuracy of our GNSS VSM retrievals using a scaled soil wetness index in the revised manuscript (see Response 1.17). This aspect was clarified in the revised manuscript (P. 8, L. 8-19; P. 10, L. 12-13 and 22; P. 11, L. 2 and 9-11; Table 1; Table 2; Fig. 3 and 4).

1.5 [The authors ask the question if phenological stages can be inferred from GNSS. The outcome and visibility of the paper could be increased by giving more specific information about different stages or managements, e.g. in form of an index or threshold for wheat as an important representative for all cereals.]

**Response 1.5:**

We found in our case study, that the tillering date (12 March) obtained from a GDD model is close to the start date (10 March) of a multiple peak period (see Section 5.5), when the vegetation height is about 20 cm, close to one wavelength. Flowering and ripening occur towards the end of the growing period when the vegetation height is no longer increased compared with 15 days before but slightly declines due to wheat heads tipping down. In order to confirm these findings, it could be recommended to perform GNSS-R measurements further over wheat fields and other crops, together with phenological stages observations. We clarified this in the revised manuscript (P. 16, L. 23-27).

**1.6 [Specific comments: Abstract: More information about the retrieval method should be added.]**

**Response 1.6:**

Soil moisture is retrieved from the multipath phase assuming the relative antenna height is constant, and the vegetation height is retrieved using the SNR's dominant period derived from a wavelet analysis. We rephrased the abstract accordingly (P. 1, L. 14-18).

**1.7 [P. 2, L. 10f: Refer also to the other L-band satellite SMAP.]**

**Response 1.7:**

Yes, we will cite the Soil Moisture Active Passive (SMAP) mission (Chan et al., 2016), in addition to SMOS. (P. 2, L. 15-16)

Reference:

Chan S. K., Bindlish, R., O'Neill, P. E., Njoku, E., Jackson, T., Colliander, A., Chen, F., Burgin, M., Dunbar, S., Piepmeier, J., Yueh, S., Entekhabi, D., Cosh, M. H., Caldwell, T., Walker, J., Wu, X., Berg, A., Rowlandson, T., Pacheco, A., McNairn, H., Thibeault, M., Martínez-Fernández, J., González-Zamora, A., Seyfried, M., Bosch, D., Starks, P., Goodrich, D., Prueger, J., Palecki, M., Small, E. E., Zreda, M., Calvet, J.-C., Crow, W., and Kerr, Y.: Assessment of the SMAP passive soil moisture product, IEEE Trans. Geosci. Remote Sens., 54 (8), 4994 - 5007, doi:10.1109/TGRS.2016.2561938, 2016.

**1.8 [P. 3, L. 15: introduce L2C.]**

**Response 1.8:**

The SNR of L2C signal is only transmitted by the recent Block IIR-M  ("Replenishment Modernized") and IIF  ("Follow-on") GPS satellites, which is with higher power and more precise than the signal L1 C/A. We introduced L2C in the revised manuscript (P. 4, L. 22-23).

**1.9 [P. 3, L. 26: What characterizes the dominant period?]**

**Response 1.9:**

The definition of the dominant period is: the peak period of the average power spectrum from the valid SNR segment data at elevation angles ranging from 5 to 20 degrees. We clarified this in the revised manuscript (P. 1, L. 17-18; P. 9, L. 10-11).

**Response 1.10:**

PBO $H_2O$ is an initiative to translate data from the Plate Boundary Observatory (PBO) sites of the GPS network in the western United States into environmental products (Larson, 2016). (P. 3, L. 3)

1.11 [P. 5, L. 15: Start the section with explaining the aim of the calculations.]

**Response 1.11:**

Due to the motion of the GPS satellites, the path delay δ between the direct and reflected signals cause an interference pattern in the signal power of SNR data. The SNR frequency/period is directly affected by the perpendicular distance from the antenna to the dominant reflecting surface. Provided the reflecting surface is stable, the a priori antenna height can be used to estimate the SNR frequency. The SNR frequency is used to calculate the multipath SNR phase. Then, the SNR phase is used to estimate VSM. If the reflecting surface is changing in response to vegetation growth, vegetation height can be retrieved instead of VSM by directly estimating the dynamic SNR frequency/period with a wavelet analysis. (P. 6, L. 6-8; P. 7, L. 3-6)

1.12 [P. 6, L.10: Again, explain in one or two sentences the general concept of soil moisture retrieval before starting the details of this section.]

**Response 1.12:**

As the SNR frequency is known (Eq. (3)), it is possible to estimate the SNR amplitude and phase. Larson et al. (2008) and Larson et al. (2010) showed that phase varies linearly with near-surface VSM ($R^2$ = 0.76 to 0.90). This result was used by Chew et al. (2014) to develop an algorithm to estimate surface soil moisture (top 5 cm) over bare ground. (P. 7, L. 8-11)

1.13 [P. 7, L. 9: A discussion about the reasons and needs for omitting a soil moisture retrieval under vegetation is necessary. Why were alternative methods not used?]

**Response 1.13:**

In conditions of significant vegetation effects, Chew et al. proposed an algorithm able to correct the phase for vegetation effects. Firstly, $A_{LSPnorm}$ and $\Delta H_{eff}$ are derived by a Lomb-Scargle Periodogram (LSP) method. Then the observed SNR metrics ($A_{norm}$, $A_{LSPnorm}$ and $\Delta H_{eff}$) are smoothed using a low-pass filter (Savitzky-Golay filter or moving average filter). A

linear nearest neighbor search algorithm with $A_{norm}$, $A_{LSPnorm}$ and $\Delta H_{eff}$ is used to find the estimated phase ($\varphi_{veg}$) caused by vegetation in a modeled lookup table. The $\varphi_{veg}$ values derived from the lookup table are then smoothed through time using the same filter. Then the expected phase changes ($\varphi_{VSM}$) due to soil moisture is equal to $\varphi_{VSM} = \Delta\varphi - \varphi_{veg}$, where $\Delta\varphi$ is the original observed phase change. This algorithm is based on the assumption that the total phase change is a linear combination of the phase change due to soil moisture and of the phase change due to vegetation. Another important difference for retrieving soil moisture with significant vegetation effects is that the slope ($S$) of the relationship between phase ($\varphi_{VSM}$) and soil moisture changes throughout the year. $S$ is a function of time, which also needs to be searched for in the lookup table. Additionally, this algorithm is based on an unpublished lookup table for new L2C GPS signals. Since the receiver we used could not track L2C signals and since we could not access a relevant lookup table, we were not able to correct for vegetation effects and we retrieved surface soil moisture over a period with rather sparse vegetation, from 16 January to 5 March. (P. 14, L. 20-23)

1.14 [P. 10, L. 3: how independent are the in situ data when some have been used for calibration? This needs to be clarified.]

**Response 1.14:**

With the a priori $S = 0.0148$ m$^3$m$^{-3}$degree$^{-1}$, only the minimum soil moisture observation during the time period is used as the $VSM_{resid}$. We also used the *in situ* soil moisture observations and phases from SNR data to fit the local slope: $S = 0.0033$ m$^3$m$^{-3}$degree$^{-1}$. In this situation, only ISBA simulations can be considered as independent from the GNSS retrievals. This aspect was clarified in the revised manuscript (P. 8, L. 3-7).

1.15 [P. 10, L. 11ff: The reason for larger variability in GPS daily soil moisture estimates could be found in different locations observed. During satellite overpasses the observed location moves within the larger "footprint" of the GNSS system.]

**Response 1.15:**

Yes. Larger variability in GPS sub-daily VSM estimates might originate from the different locations observed. Many local environment factors such as vegetation effects, precipitation, changes in soil roughness and soil composition, can perturb the GPS VSM estimates. During satellite overpasses the observed location changes together with the size of the footprint (the First Fresnel Zone) of the GNSS system, in relation to the antenna height and elevation angle range. It might be another cause of the sub-daily variability of VSM estimates. Additionally, issues with the SNR data of the L1 C/A signal and the receiving antenna gain pattern may also affect the VSM estimates. (P. 14, L. 19-20; P. 1 in the supplement)

1.16 [P. 11, L. 12: What is the reason for using a curve smoothing procedure? What are the reasons for the leveling effect?]

**Response 1.16:**

The possible causes of the leveling effect are discussed in Section 5: (1) the occurrence of more than one dominant reflecting surface at different heights (Sect. 5.3) and (2) rapid phenological changes in the wheat canopy triggering a response of the H retrieval (Sect. 5.5). It must be noted that absolute daily changes in H (and h), of about 1.1 cm d$^{-1}$ are fairly uniform throughout the growing period. Since h decreases when plants grow, relative changes in h tend to increase. According to Eq. 4, T behaves similarly. This means that the sensitivity of the retrieval method to changes in H is larger at the end of the growing period. This is probably why leveling is more pronounced between mid-March and mid-April than at the end of April (see Fig. 7). Leveling is less noticeable in May. (P. 15, L. 27-31)

1.17 [P. 12, L. 22ff: The authors could show the retrieval of a soil wetness index and relate it to in situ soil moisture by multiplying it to porosity (from in situ measurements or soil maps).]

**Response 1.17:**
(P. 8, L. 8-19; P. 10, L. 12-13 and 22; P. 11, L. 2 and 9-11; Table 1; Table 2; Fig. 3 and 4)
Yes. The phase time series can be normalized for each satellite track. Then the median value of the normalized phases from all available satellite tracks can be considered as the final soil wetness index ($\varphi_{index}$) for each day as shown in Fig. R1.1 (red line).

$$\varphi_{index} = \frac{\varphi - \varphi_{min}}{\varphi_{max} - \varphi_{min}} \qquad\qquad (R1.1)$$

This soil wetness index time series is linearly related with in situ observations (R$^2$ = 0.74) and ISBA simulations (R$^2$ = 0.65). Moreover, VSM can be estimated from $\varphi_{index}$

$$VSM = VSM_{obs\_min} + \varphi_{index} \cdot (VSM_{obs\_max} - VSM_{obs\_min}) \qquad\qquad (R1.2)$$

*VSM$_{obs\_min}$* and *VSM$_{obs\_max}$* are the minimum and maximum *in situ* VSM observations during the experimental time period, respectively. Figure R1.2 presents the estimated VSM from GPS soil wetness index ($\varphi_{index}$), together with *in situ* VSM observations and ISBA simulations. More related scores are shown in Table R1.1 and the scatter plot between GPS retrievals from $\varphi_{index}$ and *in situ* observations are shown in Fig. R1.3. We will present these results in the revised manuscript.

[Figure]

**Fig. R1.1** - Median of the daily GPS normalized phases (soil wetness index, red line) and their daily statistical distribution (black box plots) for all available satellite tracks from 16 January to 5 March 2015.

[Figure]

**Fig. R1.2** - In situ daily mean surface volumetric soil moisture (VSM) observations at 5 cm depth (green line), ISBA daily mean simulations (blue line), median of the daily GPS retrievals with soil wetness index (red line) and their daily statistical distribution (black box plots) for all available satellite tracks from 16 January to 5 March 2015.

[Figure]

**Fig. R1.3** - Scatter plot between GPS retrievals (Eq. (R1.1)) and *in situ* VSM observations $(m^3 m^{-3})$ from 16 January to 5 March 2015.

**Table R1.1** - Soil moisture scores from 16 January to 5 March 2015

| | GPS vs. *in situ* | GPS vs. ISBA | GPS vs. *in situ* | GPS vs. ISBA | GPS ($\varphi_{index}$) vs. *in situ* | GPS ($\varphi_{index}$) vs. ISBA | ISBA vs. *in situ* |
|---|---|---|---|---|---|---|---|
| S (m$^3$m$^{-3}$deg$^{-1}$) | 0.0148 | | 0.0033 | | - | - | - |
| N | 47 | 43 | 47 | 43 | 47 | 43 | 43 |
| MAE (m$^3$m$^{-3}$) | 0.036 | 0.034 | 0.011 | 0.018 | 0.007 | 0.009 | 0.009 |
| RMSE (m$^3$m$^{-3}$) | 0.046 | 0.041 | 0.014 | 0.022 | 0.009 | 0.012 | 0.010 |
| SDD (m$^3$m$^{-3}$) | 0.036 | 0.037 | 0.009 | 0.012 | 0.008 | 0.011 | 0.006 |
| Mean bias (m$^3$m$^{-3}$) | 0.029 | 0.019 | -0.010 | -0.018 | 0.003 | -0.005 | 0.008 |
| R$^2$ | 0.73 | 0.63 | 0.73 | 0.63 | 0.74 | 0.65 | 0.88 |

**Zhang et al.: Use of GNSS SNR data to retrieve soil moisture and vegetation variables over a wheat crop, Hydrol. Earth Syst. Sci. Discuss., doi:10.5194/hess-2017-152, 2017.**

**RESPONSE TO REVIEWER #2**

The authors thank anonymous reviewer 2 for his/her review of the manuscript and for the fruitful comments.

**The revised text is shown in blue in the enclosed version of the manuscript.**

2.1 [General comments
This paper presents a case study applying GNSS signals, which were reflected on the ground surface (soil, vegetation surface) to derive soil moisture and vegetation height data over a wheat crop field. The GPS antenna was installed at a height of 2.51 m. Soil moisture was retrieved as long as the vegetation height was lower than ~20 cm. However, with a further increase in plant height, it was not possible to retrieve soil moisture. Reaching a certain plant height, it was then possible to retrieve the vegetation height from the GNSS signals.
In general, the topic of this manuscript is interesting and worth to be published in HESS. The methods seem valid and transparent. However, before publishing, this manuscript has to undergo major revision as several points have to be clarified and described / discussed better / more clearly. The manuscript should undergo an English spell check. The following points should be improved in general:
- Please highlight in a more prominent way what is really new and what is the outcome and applicability of this approach.]

**Response 2.1:**

Yes. We revised the abstract and conclusions to highlight the new results presented in this study. (P. 1, L. 14-19; P. 17, L. 26-32; P. 18, L. 2-6 and 12)
In particular, the following information was given:

GNSS SNR data were obtained using the GNSS-IR technique over an intensively cultivated wheat field in southwestern France. The data were used to retrieve either soil moisture or vegetation height during the growing period of wheat. Vegetation growth tended to decrease the relative antenna height and broke up the constant height assumption used in soil moisture retrieval algorithms. Soil moisture could not be retrieved after wheat tillering. A new algorithm based on a wavelet analysis was implemented and used to extract the dominant period of the SNR and further to retrieve vegetation height.

Should a revised version of this paper be accepted in HESS, a copy editing work will be performed.

2.2 [- Please introduce and explain the so called 'dominant period' in more detail.]

**Response 2.2:**

A vegetation height retrieval algorithm is proposed using the dominant SNR period, which is the peak period in the average power spectrum derived from a wavelet analysis of SNR. We clarified this in the revised manuscript (P. 1, L. 17-18; P. 9, L. 10-11).

2.3 [- Please clarify that the GNSS retrieval of soil moisture and / or vegetation variables, actually only vegetation height, is only valid for different temporal stages. Especially, at the beginning it is unclear / confusing that soil moisture and vegetation height were retrieved at different time periods (before and after vegetation significant growth in March)].

**Response 2.3:**

Yes. We replaced "vegetation variables" or "vegetation characteristics" by "vegetation height". We mentioned in the Abstract that soil moisture and vegetation height were retrieved at different time periods (before and after vegetation significant growth in March). (P. 1, L. 18-19; P. 1, L. 12 and 29; P. 4, L. 6; P. 5, L. 17)

2.4 [- If the title contains 'vegetation variables' but only 'vegetation height' is retrieved, please change this in the title and at relevant parts of the manuscript.]

**Response 2.4:**

Yes. We modified the title as 'Use of reflected GNSS SNR data to retrieve either soil moisture or vegetation height over a wheat crop'. We replaced "vegetation variables" or "vegetation characteristics" by "vegetation height" in the entire manuscript (P. 1, L. 1-2; P. 1, L. 12 and 29; P. 4, L. 6; P. 5, L. 17).

2.5 [- The structure of the paper is not always clear – especially the chapters 'Method', 'Results' and 'Discussion' should be structured better. Some results / discussions already appear in the methods part, some points of the discussion in the results part and some methods in the discussion part.]

**Response 2.5:**

Yes. We revised the manuscript accordingly. In particular, description of Fig. 1 was moved from Sections 2.1 and 3.2 to Section 4 (P. 11, L. 25-31; P. 12, L. 1-8). Description of Fig. 2 was moved from Sections 3.1 and 3.2 to Section 4 (P. 12, L. 21-25). Description of Figs. S3 and S4 was moved from Section 3.2 to Section 4 (P. 12, L. 9-21 and 26-34; P. 13, L. 1-3).

2.6 [- In some parts, the methods are explained very well, but in some parts they are presented too extensively. The manuscript should be more focused on your applied method and should be shortened as many aspects are already published in literature and don't have to be repeated in this manuscript.]

**Response 2.6:**

Yes. We improved the focus of Section 3. Note however that all readers of HESS are not familiar with GNSS reflectometry and that Eqs. 1-6 need to be presented. We moved Eqs. 10-12 in the manuscript to the revised supplement (Eqs. S9-S11).

2.7 [- Is it necessary to retrieve soil moisture before retrieving vegetation height? Please comment on this.]

**Response 2.7:**

No. It is not necessary to retrieve soil moisture before retrieving vegetation height. This was made clear in the revised manuscript (P. 9, L. 26).

2.8 [- Regarding the statistics, 7 or even only 5 (during the period you used to demonstrate vegetation height) in situ vegetation height samples are actually too low. Please comment at least that during further studies more in situ data should be carried out.]

**Response 2.8:**

Yes. The in situ vegetation height samples are few, but it must be noted that GNSS height retrievals are totally independent from the in situ measurements. We will make clear that in further studies, more in situ data enabling the characterization of vegetation would be needed. (P. 17, L. 13)

2.9 [- It is questionable if all information given in the supplement is needed. On the other hand, some figures (see specific comments below) would also be valuable within the manuscript itself and should be presented there.]

**Response 2.9:**

Yes. We adjusted this in the revised manuscript and supplement. Fig. S3 and Fig. S8 were combined together to get a new Fig. 6 in the revised manuscript. Fig. S4 and Fig. S6 were moved to the revised manuscript as Fig. 5 and Fig. 8, respectively. Fig. S7 replaced Fig. 2 in the revised manuscript. Fig. S9 was megred into Fig. 9 in the revised manuscript. Fig. S10 was omitted becasue it was similar as Fig. 8 in the revised manuscript.

2.10 [Specific comments
Page 1 – Title
Please clarify that the retrieval of soil moisture and vegetation variables are actually only valid for different temporal stages (before and after vegetation significant growth in March).

Moreover, it would be valuable to include that you use reflected GNSS signals in your approach as also other GNSS approaches exist on this topic.
Title suggestion: 'Use of reflected GNSS SNR data to retrieve either soil moisture or vegetation height, depending on the vegetation phase of a wheat crop field,']

**Response 2.10:**

Yes, we changed the title accordingly: 'Use of reflected GNSS SNR data to retrieve either soil moisture or vegetation height over a wheat crop' (P. 1, L. 1-2)

2.11 [Page 1 – Abstract
General: The absolute length of the abstract seems fine, however, the information given here should be compressed or information should be combined more functionally. Additionally, it should be added why this approach is generally useful (1 sentence) and what is missing so far regarding the state of art (1 sentence)]

**Response 2.11:**

Yes. Surface soil moisture can be retrieved based on the linear relationship between in situ soil moisture observations and SNR phases estimated by the Least Square Estimation method, assuming the relative antenna height is constant. However, it is found in this study that the vegetation growth breaks up the constant relative antenna height assumption, and modulates the SNR period. A vegetation height retrieval algorithm is proposed using the SNR dominant period, which is the peak period in the average power spectrum derived from a wavelet analysis of SNR.
We rephrased the abstract accordingly (P. 1, L. 14-19).

2.12 [p.1, l.15: '...numerical simulations of biomass...']

**Response 2.12:**

Yes. The sentence was modified (P. 1, L. 20-21) as:

"The retrievals are compared with two independent reference datasets: *in situ* observations of soil moisture and vegetation height, and numerical simulations of soil moisture, vegetation height and above-ground dry biomass from the ISBA (Interactions between Soil, Biosphere and Atmosphere) land surface model."

2.13 [p.1, l. 18: describe in few words the 'dominant period']

**Response 2.13:**

A vegetation height retrieval algorithm is proposed using the dominant SNR period, which is the peak period in the average power spectrum derived from a wavelet analysis of SNR. We clarified this in the revised manuscript (P. 1, L. 17-18).

2.14 [p.1, l. 18: '...SNR data, whereas changes in...']

**Response 2.14:**

Yes. The sentence was rephrased accordingly (P. 1, L. 23).

2.15 [p.1, l. 20: '...smaller than one wavelength (~19 cm).' This should also be changed in the entire manuscript.]

**Response 2.15:**

Yes. We corrected it (P. 1, L. 25).
"Surface volumetric soil moisture can be estimated ($R^2 = 0.74$, RMSE = 0.009 $m^3m^{-3}$) when the wheat is smaller than one wavelength (~ 19 cm)."

2.16 [p.1, l. 22: dry biomass?]

**Response 2.16:**

Yes. We corrected this (P. 1, L. 28 and 31).

2.17 [Page 1-3: 1. Introduction
General: The introduction is quite good, but it should be written more comprehensively, especially the parts where you describe already published techniques. However, the first part (p.1, l.27-p.2, l.2) where you introduce the necessity of this approach and the recent lack to monitor land surface variables at a local scale should be extended! Moreover, it should be written more clearly why GNSS reflectometry could be a solution.]

**Response 2.17:**

In situ VSM observations are not widespread in France and in situ vegetation height observations are generally not available. Therefore, ISBA (Interactions between Soil, Biosphere and Atmosphere) simulations are key for water resource monitoring at the country scale. It must be noted that the ISBA model is forced by the SAFRAN atmospheric analysis and that SAFRAN is able to integrate thousands of in situ raingage observations. ISBA is also able to simulate vegetation characteristics such as vegetation height, leaf area index, and above-ground dry biomass. However, in situ VSM observations are needed to validate land surface models and/or satellite-derived products (e.g. Albergel et al., 2010). From this point of view, the spatial resolution of GNSS retrievals is an asset. The area sampled by GNSS retrievals is much larger than what can be achieved using individual soil moisture probes and much smaller than pixel size of satellite-derived products. Longer time periods of GNSS retrievals should be envisaged to serve as independent validation data sources in statistical methods such as Triple Collocation (Dorigo et al., 2010).
We clarified it in the revised manuscript (P. 2, L. 6-7). We also discussed it in Sect. 4.6 (P. 16, L. 29-30; P. 17, L. 1-8).

**Response 2.41:**

We moved Section 3.3 to the revised Supplement (P. 9 in the revised supplement).

2.42 [p.21, Fig. 2: Should/could be combined with Fig. S7. Figure S3 and S4: Especially Fig. S4 is interesting. It should be demonstrated within the manuscript as it shows at which stages it is difficult to retrieve the results according to the dominant period.]

**Response 2.42:**

Yes. Fig. 2 was replaced by Fig. S7. Fig. S3 and Fig. S8 were combined as the new Fig. 6 in the revised manuscript (P. 15, L. 6-8). And we moved Fig. S4 from the supplement to the revised manuscript (Fig. 5; P. 12, L. 9-25).
After 10 March, wheat height exceeded one wavelength (> 0.19 m). In addition to lower $A_{norm}$ values, an increasing number of unsuitable tracks was observed till 20 March, together with low values of peak power. The vegetation gradually decreased the strength of the signal reflected from the soil surface but increased the signal reflected from vegetation, causing more than one peak. The quality of such track data was considered too poor for retrieving biophysical variables. When the vegetation surface completely replaced the soil surface as the dominant reflecting surface of the GNSS signal, a single peak period was observed again and its value increased in response to the rise of the reflecting vegetation surface. We will revise the manuscript accordingly.

2.43 [Page 10-11: Results
p.10, l.3ff: Please insert also the mean soil moisture values of each method (for the entire observation period).]

**Response 2.43:**

Yes. The mean soil moisture values during the experimental period are 0.274 $m^3m^{-3}$ for *in situ* VSM measurements, 0.281 $m^3m^{-3}$ for ISBA simulations, 0.305 $m^3m^{-3}$ for GPS retrievals with S=0.0148 $m^3m^{-3}degree^{-1}$, 0.264 $m^3m^{-3}$ for GPS retrievals with S=0.0033 $m^3m^{-3}degree^{-1}$, and 0.276 $m^3m^{-3}$ for GPS retrievals from the scaled soil wetness index. (P. 10, L. 14-16)

2.44 [p.10, l. 3-24 and p.23, Fig. 4: Is it generally possible to compare these three methods one by one? The model simulates the first 10 cm; the reference measurements record at a soil depth of 5 cm and the GPS technique observes the soil surface. Perhaps the results with a S-value of S=0.0148 are even more realistic!? Please state on this. The GPS retrieval seems to be slightly too low in this plot using a S-value of 0.0033; especially after soil freezing and at the end of the soil moisture retrieval period the correlation between GPS retrievals and observations / reference measurements is weaker.]

**Response 2.44:**
(P. 8, L. 8-19; P. 10, L. 12-13 and 22; P. 11, L. 2 and 9-11; Table 1 and 2; Fig. 3 and 4)
Yes. Chew et al. (2014) used an electrodynamic single-scattering forward model to test the empirical relationships observed in field data, showing that SNR phase is affected by soil moisture in the top 5 cm of the soil. Moreover, surface soil moisture (< 1 cm depth) exerts the strongest control. Validation VSM obervations over the top 6 cm were used in Small et al. (2016), using the same a priori S parameter value.

We checked that the top 1 cm VSM simulations by ISBA are very close to the simulations of the top 10 cm VSM. In order to keep the method as generic as possible, we didn't directly adjust the slope from the median phase value from all available satellites. This adjusted slope value is the mean of slope values obtained for satellite tracks whose phase presents a linear correlation with *in situ* soil moisture higher than 0.9. This is why VSM retrievals are slightly too low in Fig. 4. The scores confirmed the VSM retrievals with the adjusted S parameter are closer to the in situ observations at 5 cm. Furthermore, a scaled soil wetness index can be considered, instead of VSM in $m^3 m^{-3}$ (see response 2.35).
The detail method is described below:

The phase time series can be normalized for each satellite track. Then the median value of the normalized phases from all available satellite tracks can be considered as the final soil wetness index ($\varphi_{index}$) for each day as shown in Fig. R2.1 (red line):

$$\varphi_{index} = \frac{\varphi - \varphi_{min}}{\varphi_{max} - \varphi_{min}} \qquad (R2.1)$$

This soil wetness index time series is linearly related with in situ observations ($R^2 = 0.74$) and ISBA simulations ($R^2 = 0.65$). Moreover, VSM can be estimated from $\varphi_{index}$

$$VSM = VSM_{obs\_min} + \varphi_{index} \cdot (VSM_{obs\_max} - VSM_{obs\_min}) \qquad (R2.2)$$

$VSM_{obs\_min}$ and $VSM_{obs\_max}$ are the minimum and maximum *in situ* VSM observations during the experimental time period, respectively. Figure R2.2 presents the estimated VSM from GPS soil wetness index ($\varphi_{index}$), together with *in situ* VSM observations and ISBA simulations. More related scores are shown in Table R2.1 and the scatter plot between GPS retrievals from $\varphi_{index}$ and *in situ* observations are shown in Fig. R2.3. We will present these results in the revised manuscript.

[Figure]

**Figure R2.1 -** Median of the daily GPS normalized phases (soil wetness index, red line) and their daily statistical distribution (black box plots) for all available satellite tracks from 16 January to 5 March 2015.

[Figure]

**Figure R2.2 -** In situ daily mean surface volumetric soil moisture (VSM) observations at 5 cm depth (green line), ISBA daily mean simulations (blue line), median of the daily GPS retrievals with soil wetness index (red line) and their daily statistical distribution (black box plots) for all available satellite tracks from 16 January to 5 March 2015.

[Figure]

**Figure R2.3 -** Scatterplot between GPS retrievals (Eq. (R2.1)) and *in situ* VSM observations ($m^3 m^{-3}$) from 16 January to 5 March 2015.

**Table R2.1**. Soil moisture scores from 16 January to 5 March 2015.

| | GPS vs. *in situ* | GPS vs. ISBA | GPS vs. *in situ* | GPS vs. ISBA | GPS ($\varphi_{index}$) vs. *in situ* | GPS ($\varphi_{index}$) vs. ISBA | ISBA vs. *in situ* |
|---|---|---|---|---|---|---|---|
| S ($m^3 m^{-3} deg^{-1}$) | 0.0148 | | 0.0033 | | - | - | - |
| N | 47 | 43 | 47 | 43 | 47 | 43 | 43 |
| MAE ($m^3 m^{-3}$) | 0.036 | 0.034 | 0.011 | 0.018 | 0.007 | 0.009 | 0.009 |
| RMSE ($m^3 m^{-3}$) | 0.046 | 0.041 | 0.014 | 0.022 | 0.009 | 0.012 | 0.010 |
| SDD ($m^3 m^{-3}$) | 0.036 | 0.037 | 0.009 | 0.012 | 0.008 | 0.011 | 0.006 |

| Mean bias ($m^3 m^{-3}$) | 0.029 | 0.019 | -0.010 | -0.018 | 0.003 | -0.005 | 0.008 |
|---|---|---|---|---|---|---|---|
| $R^2$ | 0.73 | 0.63 | 0.73 | 0.63 | 0.74 | 0.65 | 0.88 |

2.45 [p.10, l. 14: 'a priori'

**Response 2.45:**

Yes. We corrected it (P. 10, L. 9).

2.46 [p.11, l.6: delete '(not shown)']

**Response 2.46:**

Yes. We corrected it (P. 11, L. 19).

2.47 [p.11, l.10ff: Please insert also the vegetation height determined either by GNSS or manually for each date, instead only listing the deviations.]

**Response 2.47:**

Yes. We added a table (Table 3) to include this information in the revised manuscript (P. 13, L. 7-8).

2.48 [p.11, l.12: why do you use a 21 gliding window approach? Is this really necessary? Perhaps the vegetation height levels in Figure 6 make sense (e.g. due to meteorological events and plant growth spurts)?]

**Response 2.48:**

The possible causes of the leveling effect are discussed in Section 5: (1) the occurrence of more than one dominant reflecting surface at different heights (Sect. 5.3) and (2) rapid phenological changes in the wheat canopy triggering a response of the H retrieval (Sect. 5.5). It must be noted that absolute daily changes in H (and h), of about 1.1 cm $d^{-1}$ are fairly uniform throughout the growing period. Since h decreases when plants grow, relative changes in h tend to increase. According to Eq. 4, T behaves similarly. This means that the sensitivity of the retrieval method to changes in H is larger at the end of the growing period. This is probably why leveling is more pronounced between mid-March and mid-April than at the end of April (see Fig. 7). Leveling is less noticeable in May. A moving average permits smoothing the height retrievals, and presenting a better fit to the in situ observations. (P. 15, L. 27-31)

2.49 [p.11, l.26: Please state more on the overall possibility to compare dry biomass and vegetation height. Is this really possible? Are there some references available? Please state on this more detailed.]

**Response 2.49:**

We found a linear relationship between the moving average height from GPS retrievals and the above-ground dry biomass simulated by the ISBA model from 10 March to 29 May 2015 (when the maximum vegetation height, 1 m, was measured), during the time period from tillering to flowering. The correlation coefficient between the moving height and the dry biomass, with 81 observations, was 0.996.

$$dry\_mass = 1.05 \times moving\_height - 0.19 \qquad\qquad (R2.3)$$

with with dry mass in kg m$^{-2}$ and moving_height in meter.

A similar result was obtained by Wigneron et al. (2002) over another wheat crop site (Triticum durum, cultivar prinqual) in spring 1993. Although the sowing date (19 March) was late and the crop cycle was rather short, there was still a very good linear relationship between the in situ wheat height measurements and in situ dry biomass measurements from 20 April to 11 June 1993 (when the maximum vegetation height, 1 m, was measured). The correlation coefficient with 25 observations is 0.996.

$$dry\_mass = 1.11 \times height - 0.19 \qquad\qquad (R2.4)$$

with dry mass in kg m$^{-2}$ and height in meter.

[Figure]

**Figure R2.4 -** In situ wheat canopy height measurements (25 black dots) and in situ wheat dry biomass measurements (brown dots) from 20 April to 11 June 1993 (adapted from Wigneron et al., 2002).

We clarified this in the revised manuscript (P. 13, L. 24-30) and supplement (P. 7).

Reference:

Wigneron, J.P., Chanzy, A., Calvet, J.C., Olioso, A. and Kerr, Y.: Modeling approaches to assimilating L band passive microwave observations over land surfaces. Journal of Geophysical Research: Atmospheres, 107(D14), 2002.

2.50 [p.22/23, Fig. 3/4: for better comparability, in both Figures the y-axis should have the same scale; They could also be combined in one figure with sub-figures a] and b].]

**Response 2.50:**

Yes. We modified the figures, and combined them in one figure. We also added the retrievals from scaled soil wetness index. On the other hand, using the same y-axis scale for all sub-figures is not possible, as some sub-figures become unreadable. (Fig. 3 in the revised manuscript)

2.51 [p.24, Fig. 5: How many dots are shown in this plot (N=47?)? Please add this information in the figure capture.]

**Response 2.51:**

Yes, there are 47 dots in Fig. 5. We clarified that (Fig. 4) in the revised manuscript.

2.52 [p.25, Fig 6: You don't have to repeat the legend in the figure column.]

**Response 2.52:**
OK. (Fig. 7 in the revised manuscript)

2.53 [Page 12-14: Discussion
General: The idea of asking questions is good. Please also insert a discussion section / further question on the potential future applicability and transferability (e.g. to other soils, other vegetation types, other GNSS signals etc.). What could be improved... ]

**Response 2.53:**

Yes, we added another discussion subsection about the potential future applicability and transferability of the retrieval method. (Sect. 4.6: P. 16, L. 28-30; P. 17, L. 1-24)

We successfully assessed the surface soil moisture retrieval technique over a wheat crop field, during the start of the growing period. However, the rather narrow range of surface soil moisture values during the corresponding experiment time period limited the representativeness of the obtained retrieval accuracy. Furthermore, our dataset did not include GNSS data and in situ VSM measurements for periods of bare soil. Longer periods presenting a bare soil surface should be investigated in further studies. At the same time, more in situ vegetation measurements should be carried out for further studies.
The retrieved vegetation height was based on the dominant period of the average power spectrum. The latter was derived from GPS multipath SNR data for elevation angles between 5 and 20 degrees. We only considered the dominant period variations, without accounting for instantaneous phase changes. The accuracy of the retrieved vegetation height could probably be improved considering changes in both period and phase of the multipath SNR oscillations.
In this study, only the SNR data of L1 C/A signal is used, SNR data from different wavelength (e.g., L1 C/A, L2C and L5) should also be compared or combined to survey canopy characteristics.
A linear relationship between wheat height and above-ground dry biomass was observed during the period from wheat tillering to ripening. Retrieving dry biomass is a motivation for further research because most current satellite vegetation products focus on retrieving vegetation indexes or leaf area index. The dry biomass is directly related to the wheat yield, and retrieving wheat height could have applications in crop monitoring.
In this study, only wheat is considered. Other crops should be investigated in the future. Additionally, the algorithm we proposed might also be suitable to retrieve snow depth.

2.54 [p.12, l.3ff: As important findings (regarding the discussion) are shown in Fig. S6, this figure should also be shown in the manuscript (not only the supplement). Moreover, this issue should be discussed in more detail.]

**Response 2.54:**

Yes. We added Fig. S6 to the revised manuscript (Fig. 8) and adjusted its legend and description (P. 14, L. 6-12)

We tested the relationship between the multipath phase ($\varphi_{mpi}$) in Eq. (5) and soil moisture for the whole wheat growing cycle. We found that when the vegetation effects are not significant ($A_{norm} > 0.78$), $\varphi_{mpi}$ correlates well (R = 0.92) with the *in situ* soil moisture observations (N = 47, Fig. R2.8a). During this time period, the variation of $\varphi_{mpi}$ is only about 12 degrees in relation to the change of the in situ VSM between 0.25 $m^3m^{-3}$ and 0.30 $m^3m^{-3}$. But when the vegetation effects are significant ($A_{norm} < 0.78$), $\varphi_{mpi}$ (without or with unwrapping, Fig. R2.8b and R2.8c) is no longer linear related to soil moisture. For example, when vegetation height exceeded one wavelength, $\varphi_{mpi}$ rapidly decreased from 207 degrees to 43 degrees (between 10 and 20 March). Changes in $\varphi_{mpi}$ are disconnected from ISBA simulations. This is consistent with CH15, who showed that under this situation soil moisture cannot be retrieved unless vegetation effects are corrected for.

2.55 [p.12, l.9: Why did you increase this threshold exactly to the value 0.88? Is there any reason for this value?]

**Response 2.55:**

Adjusting the $A_{norm}$ threshold from 0.78 to 0.88 permits making a distinction between harvest and post-harvest (after 30 June) $A_{norm}$ values in Fig. S7. Fig. S7 replaced Fig. 2 in the revised manuscript. (P. 11, L. 20-24)

**Response 2.56:**

Yes. We modified it as 'Can vegetation water content be inferred from the wavelet analysis?' (P. 14, L. 24)

2.57 [p.12, l.27ff: The idea that you potentially also would like to retrieve the plant water content (or even other vegetation characteristics) should already be introduced earlier in the manuscript. Then an answer to this question would make more sense in the 'Discussion' part. Do you have reference data that show a decrease in plant water content?]

**Response 2.57:**

The VWC variable is already mentioned in the Introduction (P. 3, L. 19). The idea of retrieving VWC was expressed more clearly (P. 14, L. 25).

The conclusions of this paragraph are based on destructive gravimetric measurements (not shown).

2.58 [p.13, l.22: What do you mean with STD?]

**Response 2.58:**

Yes, we mean "daily standard deviation score". We clarified it (P. 15, L. 17-18 and 25).

2.59 [p.13, l.17: The rainfall/meteorological and logging events could additionally be shown in the figure, e.g., as a subplot.]

**Response 2.59:**

The exact day when the lodging event happened is unknown, we can only infer it happened between 29 May and 18 June. The height measurements on 29 May and 18 June are 100 cm and 39 cm, respectively. We added Fig. S9 into Fig. 7 for better comparing with rainfall data. However, whether maximum STD is an indicator of lodging or not is unclear. (P. 15, L. 19; Fig. 9 in the revised manuscript)

2.60 [p.14, l.2-8: This actually belongs to the 'Method' chapter. It is a further method to compare your retrievals to a reference.]

**Response 2.60:**

Yes. We introduced the GDD model in the 'Method' chapter. (Sect. 2.6: P. 10, L. 1-4)

2.61 [Page 14-15: Conclusions
General: Give also an outlook on potential applicability of this technique.]

**Response 2.61:**

We added a summary of the new Discussion section (See Response 2.53) (Sect. 4.6: P. 16, L. 28-30; P. 17, L. 1-24)

2.62 [p.14, l.19: Please specify – is this a new algorithm you developed or do you mean at this point the algorithm of CH15 and others you applied for the wheat crop test field?]

**Response 2.62:**

A new algorithm based on a wavelet analysis was implemented for retrieving vegetation height. We clarified it in the revised manuscript (P. 17, L. 29-31).

2.63 [p.15, l.2: L5 is introduced here for the first time. It could be mentioned already earlier (e.g., in the 'Discussion').]

**Response 2.63:**

Yes. We referred to L5 in the Discussion. (See Response 2.53) (P. 17, L. 18-19)

**Response 2.64:**

Yes. We presented a picture of the GNSS antenna in the field. (See Response 2.24) (Fig. S1 in the revised supplement)

2.65 [S p.2, Fig. S2: Applying the same time scale in the x-axis of the two plots would be better for comparability or it would even be more helpful if both plots would be combined in one figure (e.g. with two different colours).]

**Response 2.65:**

Yes. We used the same time scale in the x-axis of the two plots (Fig. S2 in the revised supplement).

2.66 [S p.4, Fig. S3: a legend would be useful; it would be logical for comparison to combine Fig. S3 and Fig. S8]

**Response 2.66:**

Yes. We added a legend and combined Fig. S3 and Fig. S8 together to get a new Fig. 6 in the revised manuscript.

2.67 [S p.5, Fig. S4: see comment above; insert a legend and units if needed.]

**Response 2.67:**

Yes. We added units in this figure and moved it to the manuscript (Fig. 5 in the revised manuscript).

2.68 [S p.6, Fig. S5: see comment above; how many dots are shown in this plot (N=47?)? Please add this information in the figure capture.]

**Response 2.68:**

Yes, N=47, we clarified it in the figure capture (Fig. S5 in the revised supplement).

2.69 [S p.7, Fig. S6: please add the black dots also to the legend; regarding the blue line / dots: use either dots or lines for all of the three plots.]

**Response 2.69:**

Yes. We modified this figure and moved it to the revised manuscript (Fig. 8). (See Response 2.54).

2.70 [S p.8, Fig. S7: see comment above.]

**Response 2.70:**

Yes. Figure S7 replaced Fig. 2 in the revised manuscript.

2.71 [S p.9, Fig. S8: see comment above.]

**Response 2.71:**
Yes. We combined this figure with Fig. S3 to make a new figure in the reviesed manuscript (Fig. 6). (See Response 2.66)

2.72 [S p.9, Fig. S9: This information could visually be combined with Fig. 2 / Fig S8.]

**Response 2.72:**
We merged Fig. S9 into Fig. 9 in the revised manuscript for better comparing with rainfall data.

2.73 [S p. 11: Duveiller et al. 2011 should also be added to the references in the manuscript.]

**Response 2.73:**

We added this reference in the revised manuscript (P. 10, L. 3)

2.74 [S p. 12: Please clarify the figure capture. Was is actually meant with '...the value retrieved 15 days before, ...'? The dates of flowering and ripening should also occur in the figure or at least in the figure capture.]

**Response 2.74:**

Yes. We clarified the caption of Fig. S7 in the revised supplement and the corresponding sentence in the revised manuscript (P. 16, L. 22-23).

[revised manuscript text omitted]

Larger variability in GPS sub-daily VSM estimates might originate from the different locations observed. Many local environment factors such as vegetation effects, precipitation, changes in soil roughness and soil composition, can perturb the GPS VSM estimates. During satellite overpasses the observed location changes together with the size of the footprint (the First Fresnel Zone, FFZ) of the GNSS system, in relation to the antenna height and elevation angle range. It might be another cause of the sub-daily variability of VSM estimates. Additionally, issues with the SNR data of the L1 C/A signal and the receiving antenna gain pattern may also affect the VSM estimates. The experiment site of the GPS receiving antenna, and the corresponding specular points and FFZ areas at 5 degrees and at 20 degrees of satellite elevation angles (outer circle and inner circle, respectively) are shown in Fig. S1.

[Figure]

**Figure S1** - (a) Antenna of the GNSS site at 2.51 m above the soil surface over an experimental field covered by rainfed winter wheat in Lamasquère, France (43°29'10"N, 1°13'57"E) on 24 April 2015. (b) Locations of the GPS specular reflection points and first Fresnel zones (FFZ). This simulation is done on 21 January 2015 for satellite elevation angles ranging from 5 to 20 degrees (outer circle and inner circle, respectively).

**SNR data**

[revised manuscript text omitted]

$$\text{biomass}_{dry} = 1.05 \times \text{height}_{moving\_avg} - 0.19 \qquad\qquad (S7)$$

with $\text{biomass}_{dry}$ (the above-ground dry biomass simulations) in kg m$^{-2}$ and $\text{height}_{moving\_avg}$ (the moving average height from GPS retrievals) in meter.

A similar result was obtained by Wigneron et al. (2002) over another wheat crop site (Triticum durum, cultivar prinqual) in spring 1993 (Fig. S6). Although the sowing date (19 March) was late and the crop cycle was rather short, there was still a very good linear relationship between the in situ wheat height measurements and in situ dry biomass measurements from 20 April to 11 June 1993 (when the maximum vegetation height, 1 m, was measured). The correlation coefficient with 25 observations was 0.996.

$$\text{biomass}_{dry} = 1.11 \times \text{height} - 0.19 \qquad\qquad (S8)$$

with $\text{biomass}_{dry}$ (in situ above-ground dry biomass measurements) in kg m$^{-2}$ and height (in situ measurements) in meter.

[Figure]

**Figure S6** - In situ wheat canopy height measurements (25 black dots) and in situ wheat above-ground dry biomass measurements (brown dots) from 20 April to 11 June 1993 (adapted from Wigneron et al., 2002).

[Figure]

**Figure S7** - The difference between retrieved vegetation height at a given date and retrieved vegetation height 15 days before, from 31 January to 11 June 2015.

**Scores**

The mean absolute error (MAE) is a quantity used to measure how close retrievals are to the observations, MAE is given by

$$MAE = \frac{1}{n} \sum_{i=1}^{n} \left| VSM_i^{OBS} - VSM_i^{GPS} \right| \tag{S9}$$

$VSM^{OBS}$ represents the *in situ* VSM observations, $VSM^{GPS}$ represents the retrieved soil moisture by GPS data, n is the valid number of data.

The root mean square error (RMSE) represents the sample standard deviation of the differences between retrieved values and observed values:

$$RMSE = \sqrt{\frac{\sum_{i=1}^{n} \left( VSM_i^{OBS} - VSM_i^{GPS} \right)^2}{n}} \tag{S10}$$

The standard deviation of the difference between observations and retrievals (SDD) is

$$SDD = \sqrt{\frac{\sum_{i=1}^{n} (x_i - \bar{x})^2}{n}} \tag{S11}$$

$x_i = VSM_i^{GPS} - VSM_i^{OBS}$, $\bar{x}$ is the mean value of x.

The fraction of explained variance is represented by the squared Pearson correlation coefficient, $R^2$.